# ITLN1 modulates invasive potential and metabolic reprogramming of ovarian cancer cells in omental microenvironment

Chi-Lam Au-Yeung[1,2 ✉], Tsz-Lun Yeung[1,2], Abhinav Achreja [3], Hongyun Zhao[3], Kay-Pong Yip[4], Suet-Ying Kwan[1], Michaela Onstad[1], Jianting Sheng[5,6], Ying Zhu[5,6], Dodge L. Baluya[7], Ngai-Na Co[1], Angela Rynne-Vidal [1], Rosemarie Schmandt[1,2], Matthew L. Anderson [8], Karen H. Lu[1,2], Stephen T. C. Wong[5,6], Deepak Nagrath [3 ✉] & Samuel C. Mok [1,2 ✉]

Advanced ovarian cancer usually spreads to the omentum. However, the omental cell-derived molecular determinants modulating its progression have not been thoroughly characterized. Here, we show that circulating ITLN1 has prognostic significance in patients with advanced ovarian cancer. Further studies demonstrate that ITLN1 suppresses lactotransferrin's effect on ovarian cancer cell invasion potential and proliferation by decreasing MMP1 expression and inducing a metabolic shift in metastatic ovarian cancer cells. Additionally, ovarian cancer-bearing mice treated with ITLN1 demonstrate marked decrease in tumor growth rates. These data suggest that downregulation of mesothelial cell-derived ITLN1 in the omental tumor microenvironment facilitates ovarian cancer progression.

[1] Department of Gynecologic Oncology and Reproductive Medicine, The University of Texas MD Anderson Cancer Center, Houston, TX 77030, USA. [2] The University of Texas MD Anderson Cancer Center UTHealth Graduate School of Biomedical Sciences at Houston, Houston, TX 77030, USA. [3] Department of Biomedical Engineering, University of Michigan, Ann Arbor, MI 48109, USA. [4] Department of Molecular Pharmacology and Physiology, University of South Florida, Tampa, FL 33620, USA. [5] Department of Systems Medicine and Bioengineering, Houston Methodist Research Institute, Weill Cornell Medicine, Houston, TX 77030, USA. [6] Center for Precision Oncology, Houston Methodist Cancer Center, Houston, TX 77030, USA. [7] Department of Diagnostic Imaging-Interventional Radiology, The University of Texas MD Anderson Cancer Center, Houston, TX 77030, USA. [8] Department of Obstetrics and Gynecology, Dan L. Duncan Comprehensive Cancer Center, Baylor College of Medicine, Houston, TX 77030, USA. ✉email: cau@mdanderson.org; dnagrath@umich.edu; scmok@mdanderson.org

Epithelial ovarian cancer (OC) is the most lethal gynecologic malignancy in the United States[1]. It occurs most frequently in postmenopausal women and metastasizes preferentially to the omentum, a well-vascularized fold of peritoneal tissue that is a major site of intra-abdominal fat accumulation. These observations indicate that visceral fat, as a component of the metastatic microenvironment, may negatively influence OC risk and progression. However, the mechanisms by which omental visceral tissue promotes tumor growth and disease progression are not entirely clear. Previous studies suggest that adipose tissue may have direct, localized effects on tumor growth, metastasis, and chemosensitivity[2–4].

The omentum is covered by a layer of mesothelial cells, which provide a slippery, nonadhesive, and protective surface for the body's serous cavities and internal organs[5]. These cells also play important roles in transporting fluid and cells across the serosal cavities, presenting antigens, controlling inflammation, and repairing tissue[5]. Disseminated OC cells adhere to and invade through the mesothelial cell layer before forming metastasized tumor nodules in the omentum, where they cause ascites, bowel obstruction, and cachexia. Many studies have demonstrated that the attachment and invasion of cancer cells may be facilitated by the binding of tumor cells to mesothelial cells' hyaluronan coats and the upregulation of adhesion molecules on cancer cells occurring in response to inflammatory mediators and the exposure of the underlying extracellular matrix[6]. However, the exact role of mesothelial cells in OC cell adhesion and growth is unclear, and the molecular mechanisms by which the OC cells modify the mesothelial cells and other cell types in the omental adipose tissues to create an environment supportive of their growth and progression have not been thoroughly explored.

In this study, we seek to identify differentially expressed genes in normal and cancer-associated omental tissue. Subsequently, we determine the functional role of an adipokine ITLN1 (intelectin-1, or omentin), one of the most significantly downregulated genes in mesothelial cells covering cancer-associated omental tissue compared with that covering normal omental tissue. We also discover the molecular mechanism by which ITLN1 suppresses the malignant phenotypes of OC cells. Finally, we evaluate the feasibility and efficacy of using ITLN1 as a therapeutic agent for OC treatment.

## Results

**ITLN1 level is lower in OC-associated mesothelial cells.** Previous studies have shown that metastatic OC cells in omental adipose tissue interact with multiple cell types to create an environment favorable to tumor growth and progression[7]. We hypothesized that identification of differentially expressed genes, particularly those coding for secretory proteins in normal versus OC-associated adipose tissue, would provide insight into the molecular mechanisms by which the omental microenvironment promotes OC progression. Therefore, we performed transcriptome profiling on microdissected omental adipose tissue from patients with benign gynecologic diseases and on cancer-associated omental tissue from patients with advanced-stage, high-grade serous ovarian cancer (HGSC). The results showed that, of the genes coding for secretory proteins, *ITLN1* was the most downregulated in omental adipose tissues from HGSC patients compared with those from patients with benign gynecologic diseases (Fig. 1a), suggesting that OC cells downregulated *ITLN1* in omental adipose tissue. After analyzing RNA-sequencing data generated from normal mesothelial cells isolated from the omental adipose tissue of healthy women, and from ascites-derived mesothelial cells isolated from HGSC patients[8], we found that *ITLN1* is the most downregulated gene in

cancer-associated mesothelial cells compared with normal mesothelial cells (Fig. 1b, Supplementary Table 1).

Based on these findings and on the fact that mesothelial cells are one of the major components of omental tissue, we hypothesized that the high levels of ITLN1 identified in normal omental adipose tissue are predominantly produced by mesothelial cells and that OC cells downregulate ITLN1 in cancer-associated mesothelial cells in the omental microenvironment. Accordingly, we performed quantitative reverse transcription polymerase chain reaction (qRT-PCR) and western blot analyses on mesothelial cells isolated from healthy women and from patients with HGSC. We found high levels of ITLN1 mRNA and protein expression in the primary mesothelial cells from healthy women, but these levels were markedly decreased in the primary mesothelial cells from HGSC patients (Fig. 1c, d). In addition, immunolocalization of ITLN1 confirmed that only calretinin-positive mesothelial cells covering the normal omental adipose tissue expressed high levels of ITLN1 and that the ITLN1 protein expression level was markedly decreased in calretinin-positive mesothelial cells covering OC-associated adipose tissue (Fig. 1e).

These data suggested that, among the cell types in the omentum, only mesothelial cells express high levels of endogenous ITLN1, and that downregulation of ITLN1 in mesothelial cells is associated with the presence of OC cells or mediators secreted by cells in the omental microenvironment, which is confirmed by co-culture experiments (Fig. 1f and Supplementary Fig. 1a).

Because OC cells have been associated with a proinflammatory microenvironment, which contain various types of stromal cells producing high levels of TNF-α and TGF-β[9], we determined whether TNF-α downregulates ITLN1 directly. We found significantly lower *ITLN1* mRNA expressions in cells treated with TNF-α compared with untreated controls (Supplementary Fig. 1b). In addition, we found markedly lower *ITLN1* mRNA expressions in TGF-β-treated mesothelial cells (Supplementary Fig. 1c). Furthermore, there was an increase in the mean circulating levels of TNF-α and TGF-β in HGSC patients compared with those in healthy women, although the change did not reach significance (Supplementary Fig. 1d, e). The expression levels of TNF-α and TGF-β receptors were also found upregulated in cancer-associated mesothelial cells compared with normal mesothelial cells (Supplementary Table 2). Collectively, our results demonstrate that key proinflammatory cytokines in the omental microenvironment downregulate ITLN1 expression in mesothelial cells in HGSC patients.

**Circulating ITLN1 levels predict overall survival rates.** Since ITLN1 is a secretory protein, we investigated whether circulating ITLN1 levels are lower in patients with HGSC than in women without cancer. We determined circulating ITLN1 levels in serum samples obtained from healthy women, as well as preoperative serum samples from patients with HGSC and from those with benign gynecologic diseases. We found that circulating ITLN1 levels were significantly lower in patients with HGSC than in healthy women or those with benign gynecologic diseases (Fig. 2a), suggesting that the presence of OC cells can lower circulating ITLN1 levels. This is further confirmed by the observation that mice injected with mouse OC cell line IG10 had significantly lower ITLN1 levels than in control mice (Fig. 2b).

Because levels of circulating CA125 protein are elevated in OC patients[10], we investigated whether there was any correlation between circulating CA125 and ITLN1 levels in patients in our cohort. We found that CA125 levels were significantly higher in patients with HGSC than in normal women or those with benign gynecologic diseases (Fig. 2c). We also observed a significant

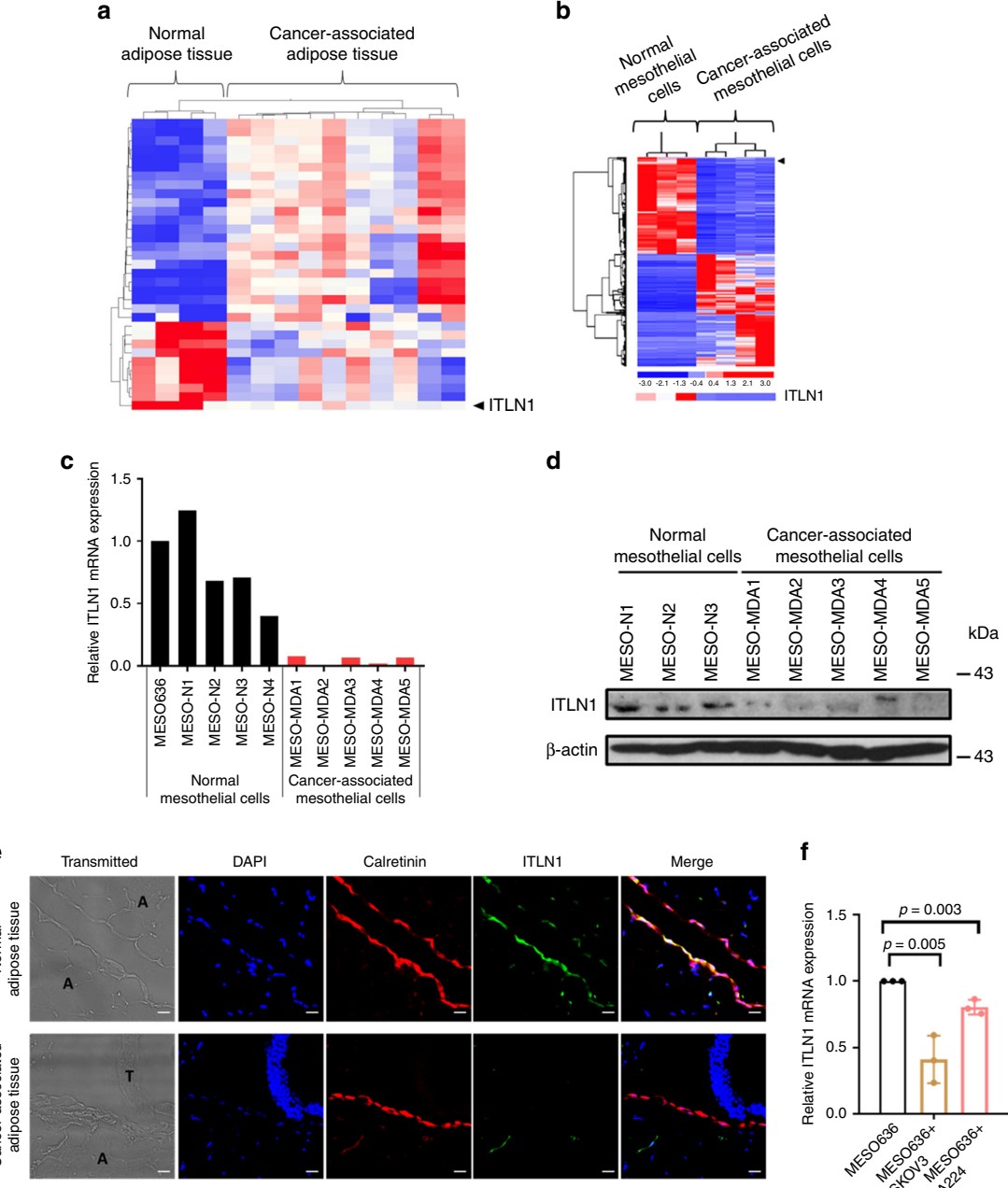

**Fig. 1 ITLN1 expression is downregulated in ovarian cancer-associated mesothelial cells.** Heat map obtained (**a**) using transcriptome profiling analysis shows *ITLN1* expression is significantly decreased in microdissected ovarian cancer-associated omental adipose tissue samples from HGSC patients ($n = 10$) compared with normal omental adipose tissue samples from patients with benign gynecologic diseases ($n = 4$); **b** using RNA-sequencing technique shows *ITLN1* expression is significantly decreased in ovarian cancer-associated mesothelial cells derived from ascites of HGSC patients ($n = 4$) compared with commercially purchased, normal mesothelial cells ($n = 3$). **c** QRT-PCR and **d** western blot analyses show a lower ITLN1 mRNA and protein expressions in ovarian cancer-associated mesothelial cells ($n = 5$) compared with normal mesothelial cells ($n = 5$ and $n = 3$, respectively). β-actin served as a loading control. **e** Representative microscopic images from immunolocalization show a lower ITLN1 expression level in the mesothelial cell layer (calretinin-positive) covering the ovarian cancer-associated omental adipose tissues compared with normal adipose tissues from healthy women. A adipocytes, T tumors; Bar = 40μm; $n = 3$. **f** QRT-PCR analysis shows a lower *ITLN1* mRNA level in MESO636 co-cultured with SKOV3 and A224 compared with MESO636 cultured alone. Results from three independent experiments were averaged and are shown as mean ± SD (two-tailed *t*-test).

inverse correlation between circulating CA125 and ITLN1 levels ($r = -0.394$; $p < 0.001$) (Fig. 2d). To determine the feasibility of using ITLN1 levels alone or in combination with CA125 levels to detect OC, we used logistic regression methods to construct receiver operating characteristic curves for ITLN1, CA125, or both to test for discriminatory ability between healthy women and women with OC (Fig. 2e). We found that CA125 alone had a significantly larger area under the curve (AUC) than ITLN1 alone

($p = 0.0031$), and that CA125 with ITLN1 had a significantly larger AUC than did CA125 alone ($p = 0.0295$) or ITLN1 alone ($p = 5.095e-6$). These data suggest that ITLN1 complements CA125 in the identification of OC patients. The prognostic significance was demonstrated by Cox proportional hazards regression model (HR, 0.65; 95% CI, 0.51–0.82; $p = 0.0004$), and Kaplan–Meier analysis using 350 ng mL$^{-1}$ as the cutoff (Fig. 2f).

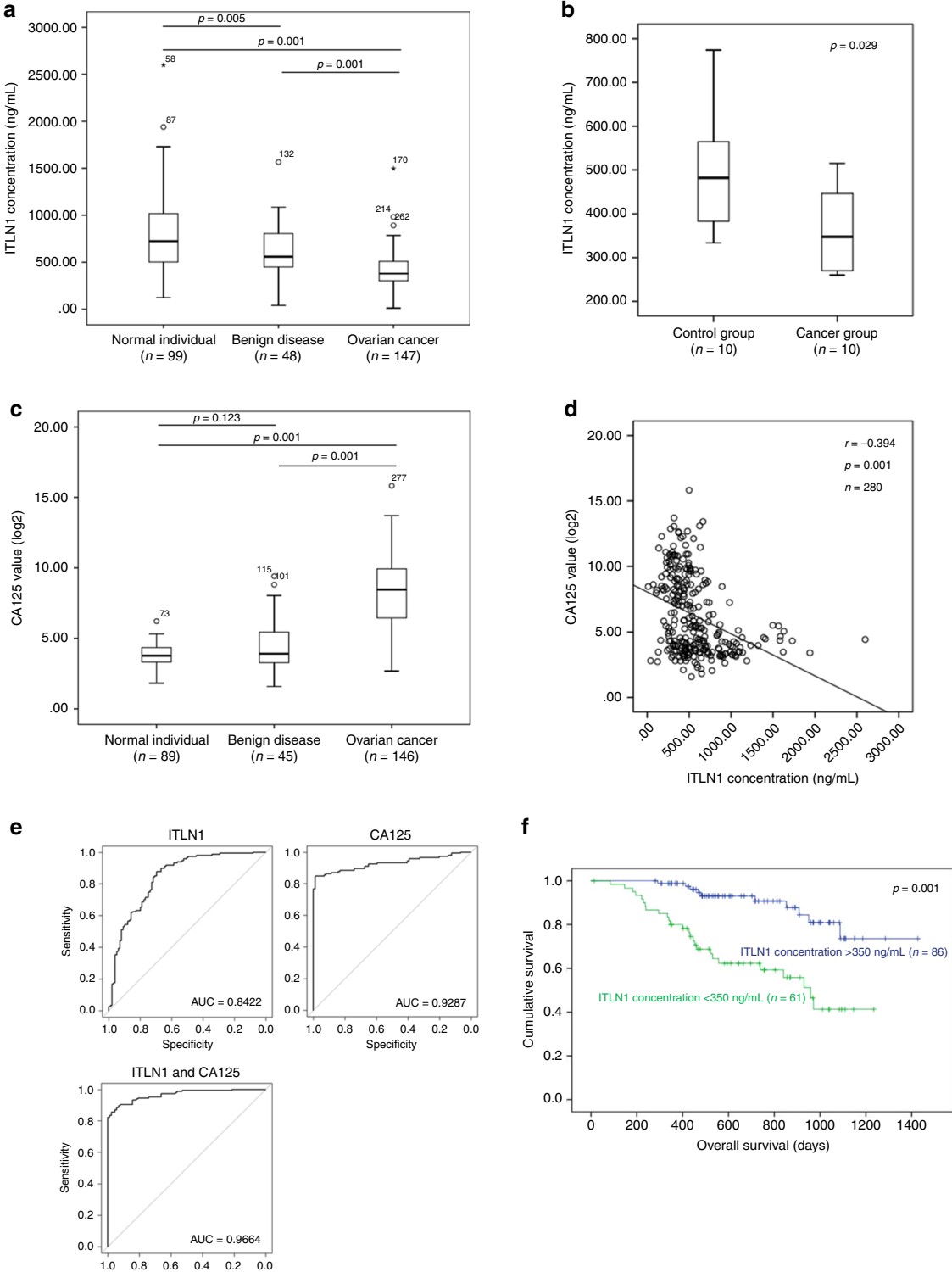

**Exogenous ITLN1 suppresses OC cells' motility**. To evaluate ITLN1's role in OC progression, we determined the effect of physiological levels of ITLN1 (500 ng mL$^{-1}$) on OC cell motility and invasive potential. Compared with SKOV3 and A224 treated with phosphate-buffered saline (PBS), cells treated with ITLN1 showed a significant decrease in the distance traveled (Fig. 3a). In addition, SKOV3 and A224 treated with ITLN1 showed significantly less invasion through a type 1 collagen-coated invasion chamber than did PBS (Fig. 3b), suggesting that exogenous ITLN1 inhibits OC metastasis.

To determine the molecular mechanism by which ITLN1 suppresses OC cells' motility and invasive potential, we performed transcriptome profiling on A224 treated with exogenous ITLN1 (Fig. 3c). To uncover the biological functions of the ITLN1-induced gene expression profile associated with cell motility and invasive potential, we used ingenuity pathway analysis to analyze the list of differentially expressed genes in A224 treated with exogenous ITLN1. One predicted activated biological function associated with cell motility and invasive potential was identified (activation $Z$-score $= -1.585$; $p = 4.92E-05$). From the list of

**Fig. 2 Higher circulating ITLN1 levels predict improved overall survival rates. a** Box plot shows a significantly lower ITLN1 level in serum collected from HGSC patients ($n = 147$) compared with normal women ($n = 99$) and patients with benign gynecologic disease ($n = 48$) ($p = 0.001$ for both comparisons; Mann–Whitney U test). **b** Box plot shows a significantly lower ITLN1 level in serum collected from C57BL/6 mice with IG10 cells injected intraperitoneally ($n = 10$) compared with the control group without cancer cell injection ($n = 10$) ($p = 0.029$; Mann–Whitney U test). **c** Box plot shows a significantly higher CA125 level in serum collected from HGSC patients ($n = 146$) compared with normal women ($n = 89$) and patients with benign gynecologic disease ($n = 45$) ($p = 0.001$ for both comparisons; Mann–Whitney U test). **d** Graph shows a negative correlation between ITLN1 and CA125 levels in serum collected from normal women, patients with benign gynecologic disease, and HGSC patients ($n = 280$; $r$ (Spearman's rank correlation coefficient) $= -0.394$; $p = 0.001$). **e** Receiver operating characteristic curves show a significantly larger area under the curve (AUC) with a combination of ITLN1 and CA125 (AUC = 0.9664) compared with ITLN1 alone (AUC = 0.8422) and CA125 alone (AUC = 0.9287) ($p = 0.001$ and $p = 0.029$, respectively; one-tailed ROC.test function). **f** Kaplan–Meier analysis shows that a high (>350 ng mL$^{-1}$) preoperative circulating ITLN1 level was significantly associated with a longer overall survival duration ($n = 147$; $p = 0.001$; two-tailed log rank test). **a–c** In the box plot, the boxes represent the interquartile range of the records, and the lines across the boxes indicate the median value of the records. The whiskers indicate the highest and lowest values among the records that are no more than 1.5 times greater than the interquartile range.

genes associated with cell motility and invasive potential, we selected *MMP1* (interstitial collagenase), long associated with invasion and metastasis[11], for further studies. QRT-PCR and western blot analyses confirmed that MMP1 mRNA and protein were downregulated in SKOV3 and A224 treated with ITLN1 compared with those threated with PBS. (Fig. 3d, e). This suggests that MMP1 mediates ITLN's effect on OC cell motility and invasion potential.

**ITLN1 abrogates LTF's effects on OC cells' motility.** To determine the signaling network by which ITLN1 downregulates MMP1, we sought to identify molecules or receptors that interact with ITLN1. A literature search did not identify any functional receptors that bind to ITLN1; however, the membrane-bound form of ITLN1 interacts with LTF (lactotransferrin)[12]. In addition, LTF's binding to one of its receptors, LRP1 (low-intensity lipoprotein-receptor-related protein 1), transcriptionally upregulates *MMP1*[13,14]. We therefore hypothesized that ITLN1 binds to LTF, preventing LTF from binding to its receptor LRP1 on OC cells and subsequently causing downregulation of MMP1. To test this hypothesis, we performed an in vitro pull-down assay on purified LTF and recombinant ITLN1 in serum-free media (SFM). A western blot analysis using anti-ITLN1 antibody on proteins pulled down by an anti-LTF antibody showed a 34 kDa ITLN1 band (Fig. 4a), suggesting that ITLN1 binds to LTF. We further confirmed that ITLN1 interferes with LTF's binding to LRP1 on OC cells with a Duolink proximity ligation assay (Fig. 4b). LTF has been reported to be mainly found in human neutrophils[15]. We validated this observation in omental tumor tissues from HGSC patients using Opal multiplex immunohistochemistry. We found that most of the LTF signals are co-localized with CD11b and CD66b (neutrophil markers) in the omental tissue (Supplementary Fig. 2a, f), suggesting that neutrophils are the major source of LTF in the omental tumor microenvironment. Further studies showed that LTF expression is significantly upregulated in cancer-associated adipose tissues from HGSC patients than in normal adipose tissues from healthy women using immunohistochemical analysis ($n = 7$ for each group, $p = 0.001$) (Supplementary Fig. 2g). We also found that there is a trend of increasing level of LTF in sera from healthy women to HGSC patients but the change is not significant. However, the level of LTF in ascites, which is rich in neutrophils, from HGSC patients is significantly higher than that in sera from any of the group examined. (Supplementary Fig. 2h) Clinically, there is no significant correlation between circulating LTF and ITLN1 levels ($r = 0.069$; $p = 0.279$), and circulating LTF and CA125 levels ($r = 0.021$; $p = 0.738$) (Supplementary Fig. 3a, b). Nevertheless, we observed a significant larger AUC in ROC curve for LTF in combination with ITLN1 and CA125 than in

that for CA125 alone ($p = 0.005$) or CA125 with ITLN1 ($p = 0.037$) (Supplementary Fig. 3c and Fig. 2e). These data suggest that LTF in combination with ITLN1 complements CA125 in identification of OC patients.

To determine whether ITLN1's inhibitory effect on OC cells' motility and invasive potential results from opposing the stimulating effect of LTF in culture media, we treated OC cells with ITLN1 in culture media supplemented with 10% fetal bovine serum (FBS) with anti-LTF antibodies or IgG. We also treated OC cells with ITLN1 or PBS in SFM. ITLN1 did not inhibit OC cell motility in either setting (Supplementary Fig. 4a, b). Finally, we treated SKOV3 and A224 with 100 μg mL$^{-1}$ (a physiological level) of LTF in SFM. The results showed significant increases in cell motility and invasive potential; however, when ITLN1 was added to the cultures, the motility and invasion rate significantly decreased (Fig. 4c, d). This suggests that ITLN1 abrogates LTF's effect on OC motility and invasive potential, and downregulation of ITLN1 allows LTF to enhance OC cells' motility and invasive potential without opposition.

Next, we evaluated whether LTF upregulates MMP1 expression in OC cells. When SKOV3 and A224 were treated with LTF in SFM, we found a significant increase in MMP1 mRNA and protein expression in both cell lines (Fig. 4e, f). When these cells were transfected with either MMP1-specific siRNAs or control siRNA (Supplementary Fig. 4c, d) and then treated with LTF, LTF's stimulating effects on cell motility were abrogated (Supplementary Fig. 4e), suggesting that LTF's effects on OC motility and invasive potential are mediated through MMP1. Subsequently, when SKOV3 and A224 were treated with LTF with different concentrations of ITLN1 (100 and 1000 ng mL$^{-1}$), ITLN1 abrogates LTF's effect on *MMP1* expression in a dose-dependent manner (Fig. 4e), suggesting that ITLN1 attenuates LTF's promoting effect on MMP1 expression in OC cells, thus suppressing OC motility and invasive potential.

To examine the mechanism by which ITLN1 attenuates LTF's motility-promoting effect on OC cells, we examined expression levels of key intermediate signaling molecules in the LTF/LRP1/MMP1 pathway in SKOV3 and A224. We discovered that LTF-treated SKOV3 and A224 showed marked increases in the levels of signaling molecules p-ERK1/2 (T202/Y204), total Jun, and p-Jun (S73) compared with cells treated with PBS in SFM (Fig. 5a). We also found marked decreases in expression levels of p-ERK1/2, total Jun and p-Jun in cells treated with LTF in the presence of ITLN1 compared with cells treated without ITLN1 (Fig. 5b). This showed that ITLN1 could abrogate LTF's effect on the activation of these signaling molecules. The roles of LRP1 and ERK activation in mediating the effect of LTF on MMP1 were further confirmed by silencing LRP1 (Supplementary Fig. 5a, b) and adding ERK inhibitor to SKOV3 and A224, respectively (Fig. 5c and Supplementary Fig. 5c).

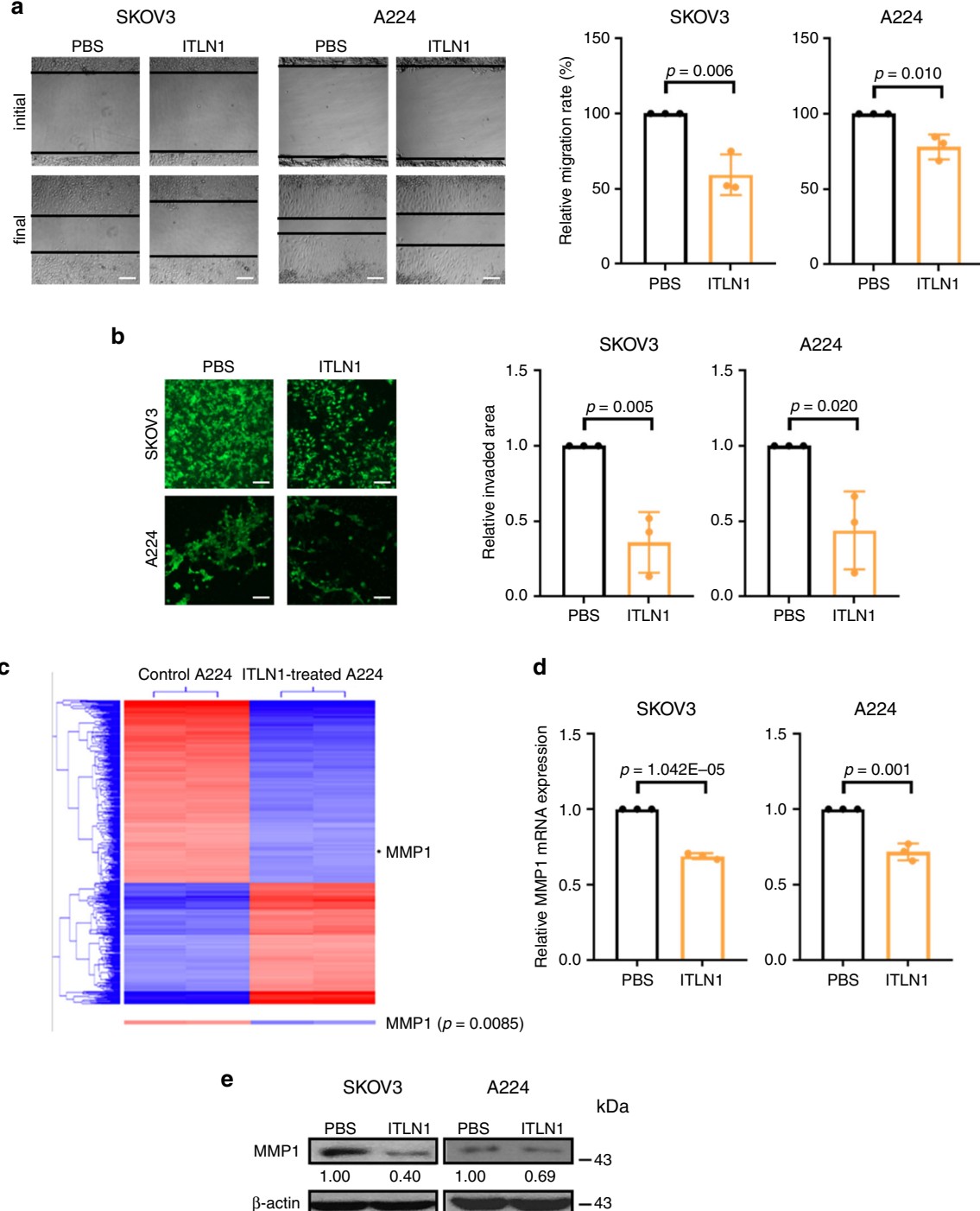

**Fig. 3 MMP1 mediates the effect of ITLN1 on suppressing ovarian cancers' motility.** Representative microscopic images of (**a**) a wound-healing assay show that ITLN1 suppressed cell migration ability in SKOV3 and A224; **b** a cell invasion assay show that ITLN1 suppressed cell invasive potential in SKOV3 and A224. Bar = 50 μm. Results in the bar charts, presented as mean ± SD (two-tailed *t*-test), show the average from three independent experiments with duplicated samples. PBS phosphate-buffered saline. **c** Heat map from a transcriptome profiling analysis shows that *MMP1* expression is significantly decreased in ITLN1-treated A224 (*n* = 2) compared with control A224 without ITLN1 treatment (*n* = 2). **d** QRT-PCR analysis shows a lower *MMP1* mRNA level in ITLN1-treated SKOV3 and A224 compared with control cells treated with PBS. Results, as presented as mean ± SD (two-tailed *t*-test), show the average from three independent experiments. **e** Western blot analysis shows a lower MMP1 protein level in ITLN1-treated SKOV3 and A224 compared with control cells treated with PBS. β-actin served as a loading control. Relative normalized protein levels with respect to the corresponding control are presented. Three independent experiments were performed.

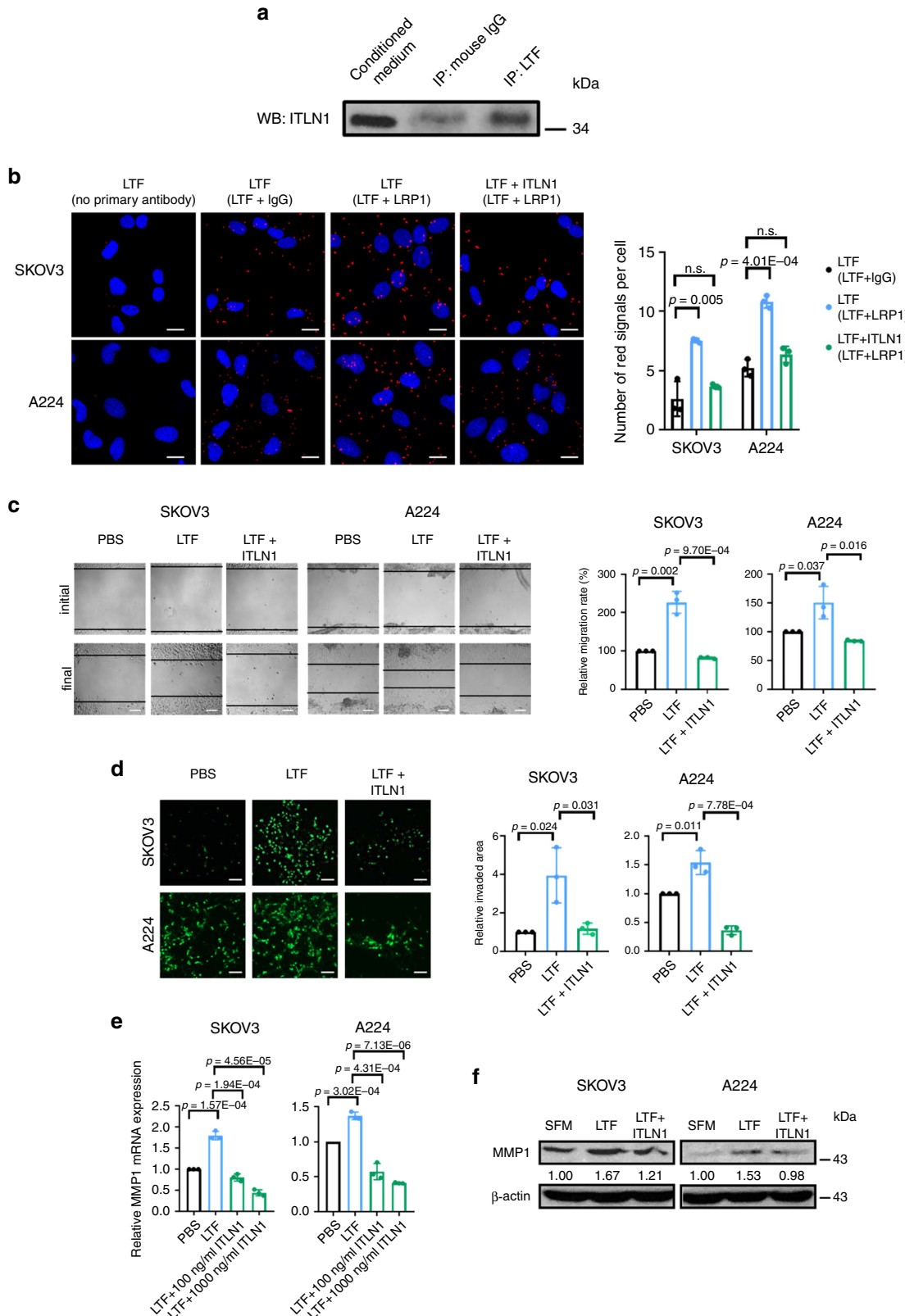

In addition to MMP upregulation, increased stress fiber formation and calcium mobilization in cells play an essential role in increasing the motility and invasive potential[16,17]. We found that LTF-treated SKOV3 and A224 showed a marked increase in stress fiber formation and intracellular calcium mobilization compared with control cells. However, addition of ITLN1 abrogated LTF's effects (Supplementary Fig. 5d, e). These observations suggest that ITLN1 attenuates LTF's motility-promoting effect by modulating calcium mobilization and stress fiber formation in OC cells.

**Fig. 4 ITLN1 abrogates LTF's effects on ovarian cancer cells' motility. a** Western blot analysis shows ITLN1 on proteins pulled down by an anti-LTF antibody. Normal mouse IgG served as a control. ITLN1 band size = 34 kDa. **b** Representative microscopic images from a Duolink proximity ligation assay (PLA) on SKOV3 and A224 shows the interaction between LTF and LRP1 while ITLN1 reduced the interaction. Red fluorescent signals indicate protein–protein interaction; nuclei were stained with 4,6-diamidino-2-phenylindole (DAPI) blue. Staining with no primary antibody and with anti-LTF antibody plus normal rabbit IgG served as controls. Bar = 5 μm. Representative microscopic images show that LTF induced **c** cell migration ability and **d** cell invasion potential in SKOV3 and A224 compared with control cells, while ITLN1 counteracted the effect. Bar = 50 μm. **e** Bar charts show that LTF upregulated the relative *MMP1* mRNA expression in SKOV3 and A224, while ITLN1 counteracted the effect. **f** Western blot analysis shows that LTF upregulated MMP1 protein expression in SKOV3 and A224, while ITLN1 counteracted the effect. β-actin served as a loading control. Relative normalized protein levels with respect to the corresponding control are presented. Three independent experiments were performed. **b**, **e** Results from three independent experiments were averaged and are shown as mean ± SD (two-tailed *t*-test). n. s. not significant ($p > 0.05$).

**Adipocytes mediate ITLN1's growth-suppressive effect on OC.** To determine ITLN1's effect on OC cell growth, we performed an MTT assay on SKOV3 and A224 treated with ITLN1. We found no significant changes in cell proliferation in ITLN1-treated cells (Supplementary Fig. 6a) compared with PBS-treated control cells, suggesting that ITLN1 has no direct effect on OC cell growth. Because ITLN1 increases insulin-dependent glucose uptake exclusively in adipocytes[18], we next asked whether ITLN1 could suppress OC cell growth in the presence of adipocytes. Mature adipocytes were characterized using Oil Red O staining (Supplementary Fig. 6b) and were co-cultured with SKOV3 or A224 in a transwell insert in the presence of ITLN1 or PBS. We found a significant decrease in growth rates in SKOV3 and A224 co-cultured with mature adipocytes and ITLN1 compared with those co-cultured with mature adipocytes alone (Fig. 6a). Similar results were observed when we repeated the experiment using two additional HGSC cell lines, OVCA432 and OVCA433 (Supplementary Fig. 6c). To determine whether ITLN1's growth-suppressive effect on OC cells is observed only in the presence of mature adipocytes, we replaced mature adipocytes with pre-adipocytes and mesothelial cells; ITLN1's growth-suppressive effect on OC cells was not observed (Supplementary Fig. 6d, e). This suggests that only mature adipocytes play a role in mediating ITLN1's growth-suppressive effect on OC cells.

We hypothesized that high ITLN1 levels may suppress ovarian tumor growth by increasing adipocytes' glucose uptake in the omental microenvironment, thereby decreasing the glucose available to neighboring OC cells. We also hypothesized that the presence of OC cells would lower local and circulating ITLN1 levels, leading to adipocytes' decreased glucose uptake, and increased local and circulating glucose levels that would fuel OC cells and support their progression. To test these hypotheses, we confirmed that ITLN1 increased insulin-dependent glucose uptake only in mature adipocytes (Fig. 6b). We then showed that adding glucose to media abrogated the growth-suppressive effect of ITLN1-treated mature adipocytes on SKOV3 and A224 (Supplementary Fig. 6f), further confirming glucose's role in mediating OC cell growth.

Next, we evaluated ITLN1's effect on the expression of GLUT4/SLC2A4 in mature adipocytes and delineate the mechanism by which ITLN1 increases glucose uptake in adipocytes. We found significantly higher levels of GLUT4 mRNA and protein expression in ITLN1-treated mature adipocytes than in controls in the presence of insulin (Fig. 6c, d), suggesting that ITLN1 upregulates GLUT4 expression in adipocytes, leading to increased glucose uptake. There is, however, no significant change of *GLUT4* mRNA expression in ITLN1-treated mature adipocytes than in controls in the absence of insulin (Supplementary Fig. 7a). We next determined the amount of glucose uptake in ITLN1-treated mature adipocytes in the presence of GLUT4-specific siRNAs or control siRNA (Supplementary Fig. 7b, c). We found significantly reduced glucose uptake in adipocytes transfected with GLUT4 siRNAs compared with those transfected with

control siRNA (Fig. 6e), suggesting that GLUT4 is essential for mediating ITLN1's effect on adipocytes' glucose uptake. To determine the role adipocytes' GLUT4 expression plays in OC growth, adipocytes transfected with GLUT4-specific or control siRNAs were co-cultured with OC cells in the presence of ITLN1. We found that ITLN1's suppression on OC cell growth was abrogated when GLUT4 was silenced in adipocytes (Fig. 6f). This suggests that ITLN1 suppresses OC growth by increasing adipocytes' glucose uptake through GLUT4 upregulation.

We next asked whether LTF played a role in decreasing glucose uptake and GLUT4 expression in adipocytes. We found that mature adipocytes treated with LTF in SFM showed a significant decrease in GLUT4 mRNA expression level and glucose uptake compared with those treated with PBS, and adding ITLN1 abrogated the effects (Fig. 6g, h). But ITLN1 has no significant effect on *GLUT4* mRNA expression in SFM (Supplementary Fig. 7d). We also demonstrated that LTF promoted cell growth in OC cells co-cultured with adipoytes in SFM and that adding ITLN1 to the co-culture abrogated LTF's growth-promoting effect (Supplementary Fig. 7e). This suggests that decreased levels of ITLN1 allow LTF to downregulate GLUT4 and decrease glucose uptake in adipocytes without opposition, leading to enhanced tumor cell growth.

To determine whether ITLN1 plays a role in adipocyte-mediated metabolic reprogramming in OC cells, we performed a lactate secretion assay on SKOV3 and A224 co-cultured with ITLN1-treated adipocytes or adipocytes alone. We observed a significantly lower level of lactate secretion in OC cells co-cultured with ITLN1-treated adipocytes compared with those with adipocytes alone (Fig. 6i). A $^{13}C$ gas chromarography–mass spectrometry (GC–MS)-based isotope tracer analysis using U-$^{13}C_6$ glucose futher confirmed reduction of glycolytic flux in A224. We found that the percentages of $^{13}C$-glucose-derived M3 pyruvate and M3 lactate were significantly lowered in A224 co-cultured with ITLN1-treated adipocytes compared with PBS-treated adipocytes (Fig. 6j, k), suggesting that smaller amounts of glucose metabolize to pyruvate and lactate in A224 co-cultured with ITLN1-treated adipocytes, and that ITLN1 suppresses OC growth through adipocyte-mediated metabolic reprogramming in OC cells.

**ITLN1 suppresses OC progression via metabolic shift in vivo.** The effect of ITLN1 on OC progression in vivo was determined by injecting mouse OC IG10 intraperitoneally into immuno-competent C57BL/6 mice. We confirmed that recombinant mouse ITLN1 suppressed the migration rate and downregulated MMP1 protein expression in IG10 in vitro (Supplementary Fig. 8a, b). The effect was similar to that seen in human OC cells (Fig. 3a, e). We then determined the optimal intraperitoneal injection concentration of recombinant mouse ITLN1. From 0 to 6 h post injection, the 100 μg kg$^{-1}$ and 1 mg kg$^{-1}$ injections did not cause a significant change in the circulating glucose

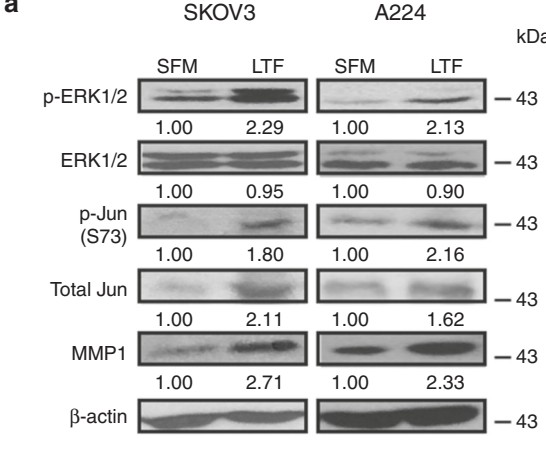

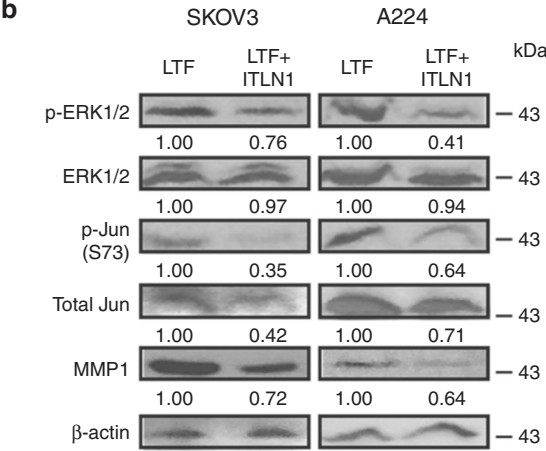

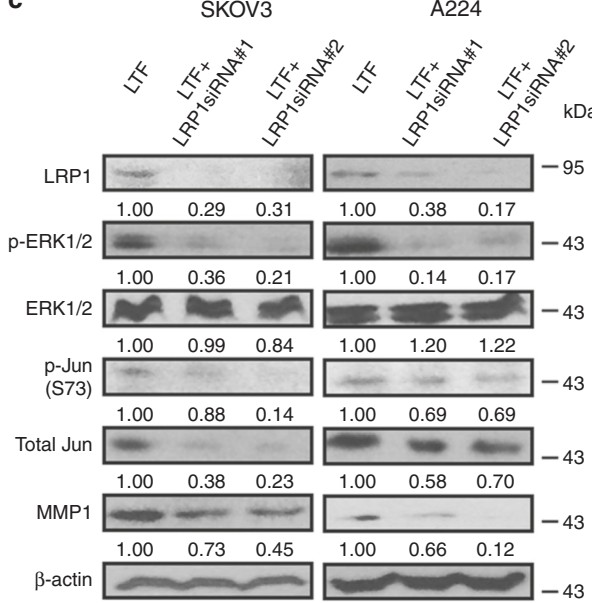

**Fig. 5 ITLN1 inhibits the LTF/LRP1/MMP1 signaling pathway.** Western blot analyses show (**a**) higher protein levels of p-ERK1/2, p-Jun (S73), total Jun, and MMP1 in LTF-treated (100 μg mL$^{-1}$) SKOV3 and A224 compared with control cells without treatment; **b** lower protein levels of p-ERK1/2, p-Jun (S73), total Jun, and MMP1 in LTF-treated SKOV3 and A224 with ITLN1 (500 ng mL$^{-1}$) compared with control cells without ITLN1; and **c** lower protein levels of LRP1, p-ERK1/2, p-Jun (S73), total Jun, and MMP1 in LTF-treated SKOV3 and A224 with LRP1-specific siRNAs compared with control cells with negative siRNA. β-actin served as a loading control. Relative normalized protein levels with respect to the corresponding control are presented. Three independent experiments were performed.

physiological ITLN1 concentration in mice without OC 48 h post injection (means: 481.75 and 495.87 ng mL$^{-1}$, respectively) (Fig. 2b and Supplementary Fig. 8d). Significant decrease in circulating glucose concentration were observed from 0 to 6 h post injection (Supplementary Fig. 8e). These data suggest that normal circulating ITLN1 levels can be achieved within 48 h with one dose of recombinant ITLN1 (2 mg kg$^{-1}$).

Next, we injected recombinant mouse ITLN1 (2 mg kg$^{-1}$) intraperitoneally into IG10-bearing C57BL/6 mice every 2 days for 6 weeks. We found a significant decrease in tumor growth rates (Fig. 7a, b), significant lower ascites volumes (Fig. 7c), and significantly higher circulating ITLN1 levels and lower glucose levels in ITLN1-treated mice compared with control mice (Fig. 7d, e). Tumor sections from ITLN-treated mice also showed significantly lower MMP1 expression levels than did sections from control buffer-treated mice (Fig. 7f).

To determine ITLN1's effect on the metabolic reprogramming of OC cells in vivo, we performed MALDI-imaging mass spectrometry (MALDI-IMS) on tissue sections from omental tumors from mice treated with one dose of ITLN1 at different time points. We found marked decreases in hexose-6-phosphate (glucose-6-phosphate and fructose-6-phosphate) in the tumor cells 1 h after ITLN1 injection. Within the same sections, hexose-6-phosphate increased in adipocytes adjacent to tumor cells (Fig. 7g, h). Lactate and ATP levels were markedly decreased in tumor cells but markedly increased in adipocytes adjacent to tumor cells (Supplementary Fig. 8f, g). This suggests that ITLN1 treatment decreases glucose uptake of metastatic OC cells but increases glucose consumption of neighboring adipocytes in the omental microenvironment. The competitive uptake of glucose by adipocytes leads to glucose-starvation of the malignant cells.

## Discussion

Our study shows that mesothelial cells in OC-associated omental tissue express significantly lower levels of ITLN1 than cells in normal omental tissue, and that circulating ITLN1 levels are significantly lower in OC patients than in healthy women and patients with benign gynecologic diseases. Functional studies revealed that ITLN1 suppresses OC's invasive potential by decreasing LTF's upregulation of MMP1 expression in OC cells; in addition, ITLN1 suppresses OC proliferation by attenuating the stimulating effect of LTF on insulin-dependent glucose uptake in adipocytes. Furthermore, we demonstrated that recombinant ITLN1 suppresses OC growth in vivo.

Metastatic cancer cells from the fallopian tube or the ovary adhere to the mesothelium, through which it invades to the visceral organs. These cells may enhance their ability to adhere and propagate by altering gene expression in mesothelial cells. When we performed an RNA-sequencing analysis on normal mesothelial cells, to our surprise, we found high levels of ITLN1 expression, although ITLN1 was originally thought to be produced by adipocytes in visceral adipose tissue[18]. Our studies

concentration in IG10-bearing C57BL/6 mice, but the 2 mg kg$^{-1}$ injection resulted in a significant decrease (Supplementary Fig. 8c). Next, we used ELISA to determine how long 2 mg kg$^{-1}$ of recombinant mouse ITLN1 injected in IG10-bearing mice would be effective. Significant increases in circulating ITLN1 concentration were observed from 0 to 48 h post injection, and the mean ITLN1 concentration became comparable with the

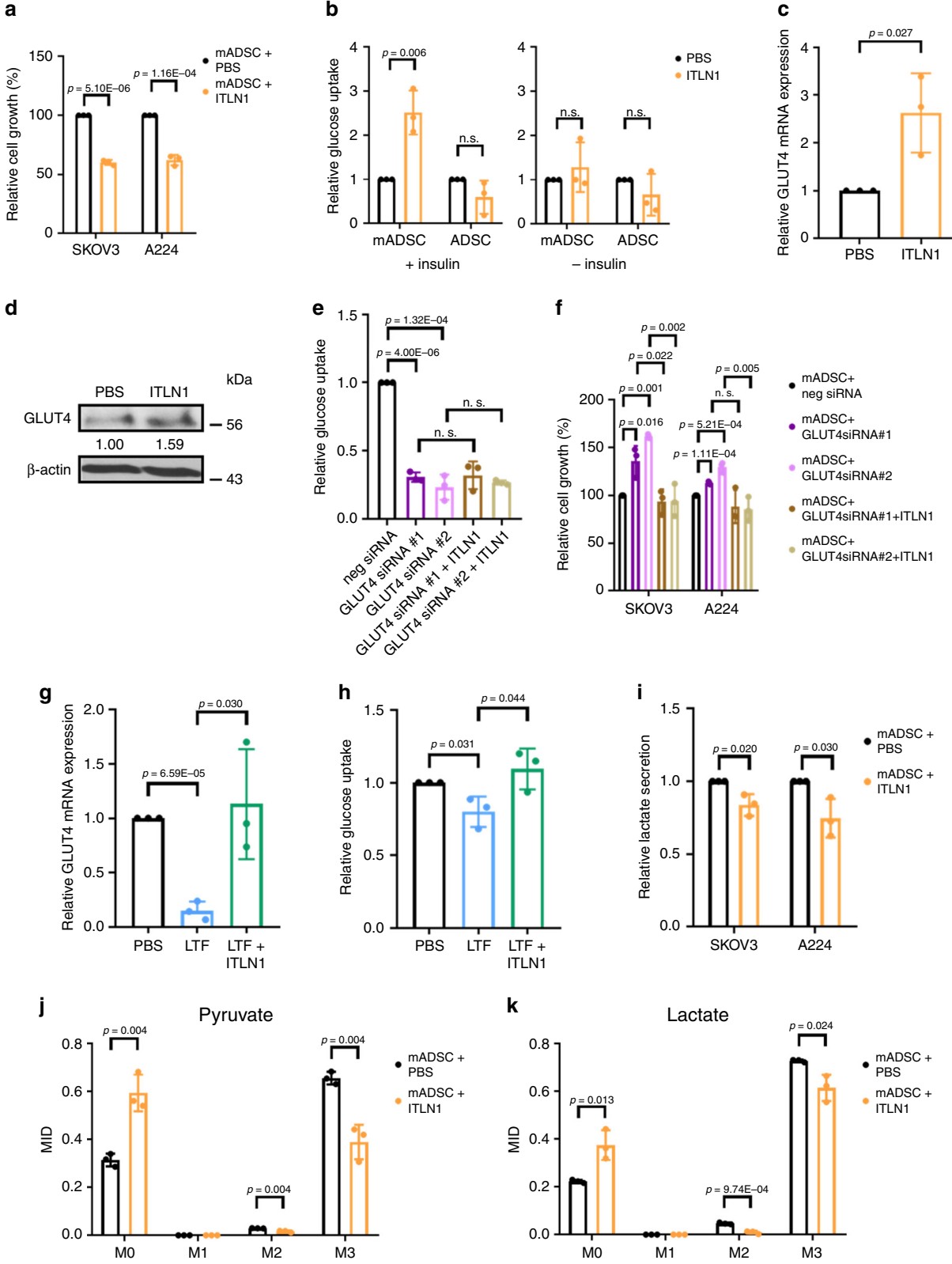

confirmed that the mesothelial cells express high levels of ITLN1. This observation can explain why ITLN1 expression was only detected in visceral adipose tissue but not in subcutaneous adipose tissue.

The plasma ITLN1 level is inversely correlated with BMI, waist circumference and insulin resistance, and it increases with weight loss[19–21]. We found that ITLN1 levels in preoperative serum samples collected from OC patients were significantly lower than those from BMI-matched patients with benign gynecologic diseases, suggesting that common mediators from the proinflammatory omental microenvironment in both obese and OC patients can suppress ITLN1 expression in mesothelial cells. Our finding that mesothelial cells treated with TNF-α and TGF-β showed a significantly lower level of ITLN1 supports our

**Fig. 6 ITLN1 suppresses ovarian cancer cell growth by enhancing glucose uptake in adipocytes. a** ITLN1 suppressed SKOV3 and A224 cell growth in the presence of mature adipocytes (mADSC). **b** ITLN1 enhanced insulin-dependent glucose uptake in mADSC but not in adipose-derived stem cells (ADSC), when both are compared with control cells without ITLN1 treatment. ITLN1 also has no significant effect on glucose uptake in mADSC and ADSC in the absence of insulin. **c** ITLN1 upregulated *GLUT4* mRNA expression in mADSC compared with control cells without ITLN1 treatment in the presence of insulin. **d** Western blot analysis shows a higher GLUT4 protein level in ITLN1-treated mADSC in the presence of insulin compared with control cells. β-actin served as a loading control. Relative normalized protein levels with respect to the corresponding control are presented. Three independent experiments were performed. GLUT4 siRNA transfection counteracted ITLN1's effect on **e** insulin-dependent glucose uptake in mADSC and **f** cell growth in SKOV3 and A224. LTF downregulated **g** the relative *GLUT4* mRNA expression and **h** insulin-dependent glucose uptake in mADSC, while ITLN1 counteracted the effects. **i** ITLN1 reduced lactate production in SKOV3 and A224 in the presence of mADSC. **j, k** GC–MS results were reported as mass isotopologue distributions (MID), which represented the relative abundance of different mass isotopologues of each metabolite; M0 referred to the isotopologues with all 12C atoms, and M1 to M3 referred to heavier isotopologues with one to three 13C atoms derived from the tracer. The (**j**) M3 pyruvate fraction and (**k**) M3 lactate fraction indicated significantly lower glycolytic fluxes in A224 co-cultured with ITLN1-treated mADSC compared with controls not treated with ITLN1. **a–c**, **e–k** Results from three independent experiments were averaged and are shown as mean ± SD (two-tailed *t*-test). n. s. not significant (*p* > 0.05).

hypothesis. Although the circulating levels of TNF-alpha and TGF-beta are not significantly higher in HGSC patients than in healthy women, the localized levels of these cytokines in mesothelial cells are increased, so do the levels of their corresponding receptors. This suggests that the expression level of ITLN1 in mesothelial cells in HGSC patients is regulated by these cytokines in an autocrine and paracrine manner in the tumor microenvironment.

Our results showed that ITLN1 downregulated MMP1 mRNA expression in OC cells cultured in 10% FBS, but had no effect in cells cultured in SFM. Downregulation of MMP1 in ITLN1-treated OC cells will therefore likely lead to deceased invasive potential. The suppressor effect can only be observed in the presence of FBS, suggesting that ITLN1 may inhibit tumor invasion by counteracting mediators in the serum. LTF is the most likely candidate. LTF is mainly found in human exocrine fluids and human neutrophils. Several studies have demonstrated the importance of neutrophils in tumor progression. An elevated neutrophil-to-lymphocyte ratio is associated with poor survival of subjects with cancer[22,23]. Binding of soluble ITLN1 to LTF prevents LTF from binding to LRP1 on the surface of OC cells, thus inactivating downstream signaling pathways that control MMP1 expression. Downregulation of ITLN1 in the proinflammatory omental microenvironment allows LTF to activate the LTF/LRP1/MMP1 pathway without opposition and to enhance the invasive potential of OC cells.

The tumor-promoting role of LTF as demonstrated in this study, is supported by a recent report that 100 μg mL$^{-1}$ LTF promotes invasiveness in triple-negative breast cancer[24]. In contrast, LTF has also been reported to have antitumor effects in breast and cervical cancers transfected with recombinant adenoviruses expressing human LTF and in head and neck cancer treated with 800 μg mL$^{-1}$ LTF[25–27]. The discrepancy is likely due to the fact that physiological LTF levels, which range from 10 μg mL$^{-1}$ in healthy individuals to 200 μg mL$^{-1}$ in individuals with inflammation[28], were not used consistently in these previous studies.

It has been shown that higher glucose levels stimulate tumor growth and progression, while reduced amounts of dietary carbohydrate or glucose can suppress tumor growth[29,30]. In the omental microenvironment, ITLN1 increases insulin-dependent glucose uptake in adipocytes, which leads to a decrease in the OC proliferation rate, as we observed in this study. It may be that, in the presence of ITLN1, cancer cells have less locally-available glucose from which they can generate energy. Our results showed that pyruvate and lactate production in OC cells were markedly lower when cells were co-cultured with adipocytes treated with ITLN1 rather than PBS. Reduced glycolysis was also observed in vivo in cancer cells in mice intraperitoneally injected with ITLN1, but increased glycolysis was observed in the adjacent

cancer-associated adipocytes. Although increased glucose uptake was observed in adipocytes from 0 to 48 h post injection, glucose-6-phosphate was also increased in tumors at later time points, perhaps because the glucose amount in mice exceeded the uptake ability of the adipocytes, and glucose in the omental microenvironment was replenished by circulating glucose when mice were treated with just one ITLN1 injection. We observed a significantly lower circulating glucose level in mice with repeated compared with no ITLN1 injections.

In conclusion, this study provides the evidence that OC cells modify mesothelial cells in visceral adipose tissue by downregulating ITLN1 to promote the invasion potential and proliferation of OC cells in the omental microenvironment. Visceral adipose tissue also puts competitive nutritional pressure on OC cells by depleting glucose through ITLN1. ITLN1 may also serve as a novel prognostic marker for OC. Finally, therapeutic strategies to upregulate ITLN1 in OC patients could inhibit OC progression and metastasis and, improve patient survival rates (Supplementary Fig. 9).

## Methods

**Cell lines and culture conditions**. Human OC lines A224 (a gift from Dr. Michael Birrer's laboratory at the Massachusetts General Hospital), SKOV3 (American Type Culture Collection, Manassas, VA, USA), OVCA432 and OVCA433 (gifts from Dr. Robert Bast's laboratory at MD Anderson Cancer Center), and mouse OC line IG10 (also from Dr. Bast's laboratory) were maintained in RPMI 1640 medium supplemented with 10% FBS, 2 mM glutamine, and penicillin–streptomycin (Life Technologies Corp., Grand Island, NY, USA). Primary normal mesothelial cells were isolated from omental adipose tissue from consented non-oncological adult donors undergoing elective gastric bypass surgery (Zen-Bio, Inc., Research Triangle Park, NC, USA) or from peritoneal washing from patients with benign gynecologic diseases[31]. They were maintained in mesothelial cell growth medium (MSO-1; Zen-Bio, Inc.). Primary cancer-associated mesothelial cells were cultured from the peritoneal effusions of HGSC patients. They were characterized using calretinin immunostaining and were maintained in 1:1 MCDB105/199 medium (Sigma-Aldrich Co., St. Louis, MO, USA) supplemented with 15% FBS, 1 ng mL$^{-1}$ epidermal growth factor, and penicillin–streptomycin (Life Technologies Corp.). Mature adipocytes (mADSC) were differentiated from adipose-derived stem cells (ADSC) (American Type Culture Collection) under three cycles of differentiation. Each of which included 3 days of incubation in DMEM medium supplemented with 10% FBS, 0.5 mM 3-isobuty-1-methylxanthine, 200 μM indomethacin, 1 μM dexamethasone, and 10 μg mL$^{-1}$ insulin (Sigma-Aldrich Co.), followed by 1 day of incubation in DMEM medium supplemented with 10% FBS and 10 μg mL$^{-1}$ insulin. The cells were characterized using Oil Red O staining and were maintained in DMEM medium supplemented with 10% FBS. All cell lines tested negative for mycoplasma contamination and were authenticated by short tandem repeat profiling in the Characterized Cell Line Core Facility at MD Anderson Cancer Center.

**Microarray analysis**. Normal adipose tissue and cancer-associated adipose tissue were derived from the omental tissues of patients with benign gynecologic diseases and HGSC, respectively. Adipose tissue was first digested with collagenase (Sigma-Aldrich Co.) in HEPES. The digested tissues were then centrifuged. The pellet containing preadipocytes, fibroblasts, and red blood cells was removed while the top fatty layer was washed in fresh DMEM/F12 medium twice, with centrifugation, before being transferred to the cell culture flasks. The flasks were fully filled with 10% FBS DMEM/F12 medium and were inverted so that the bottom of the flask

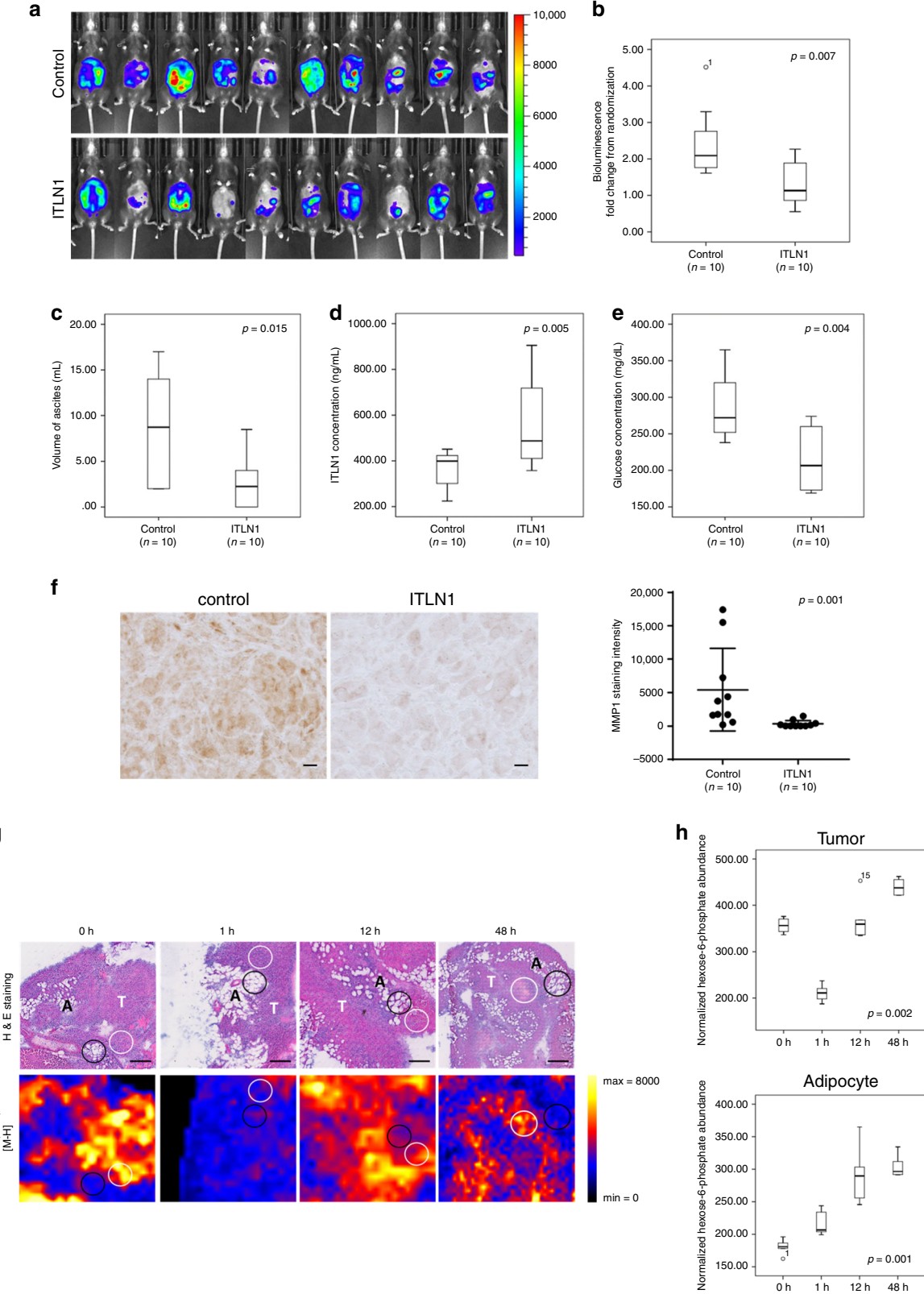

was on top. The floating mature adipocytes were attached to the upper portion of the flasks, while any fibroblast-like cells were sunk to the bottom. After 5–7 days to allow sufficient attachment of the mature adipocytes, the media inside were removed and replaced with 5 mL of fresh media. The flasks were then re-inverted for normal observation and manipulation. The mature adipocytes isolated were characterized using Oil Red O staining[32]. OC A224 cells were treated with PBS or 500 ng mL$^{-1}$ recombinant human ITLN1 for 24 h before RNA extraction. Total RNAs were extracted using TRI reagent (Molecular Research Center, Cincinnati,

OH). Purified RNA samples were amplified, labeled, and hybridized onto Affymetrix GeneChip Human Genome U133 Plus 2.0 microarrays (Affymetrix Inc., Santa Clara, CA, USA) according to the manufacturer's protocol. The arrays were then washed and stained using the Affymetrix Fluidics Station 450 (Affymetrix Inc.). Scanning of the arrays was done in the Cancer Genomics Laboratory at MD Anderson Cancer Center using a GeneChip Scanner 3000 7G (Affymetrix Inc.). A heat map was generated using dChip software (Affymetrix Inc.). Red color indicates fold change of genes >1 and blue color indicates fold change of genes <1.

**Fig. 7 ITLN1 suppresses ovarian cancer progression via metabolic shift in vivo. a** Images show a decrease in bioluminescence of IG10 cell-bearing C57BL/6 mice with ITLN1 treatment ($n = 10$) compared with untreated controls ($n = 10$) 6 weeks after treatment. Box plot shows a significantly (**b**) smaller tumor growth rate in; **c** smaller volume of ascites collected from; **d** higher ITLN1 concentration in serum collected from; and **e** lower glucose concentration in whole blood collected from IG10 cell-bearing C57BL/6 mice with ITLN1 treatment ($n = 10$) compared with untreated controls ($n = 10$) ($p = 0.007$, $p = 0.015$, $p = 0.005$, and $p = 0.004$, respectively; Mann–Whitney $U$ test) 6 weeks after treatment. **f** Representative microscopic images of paraffinized sections of tumor tissues collected from IG10 cell-bearing C57BL/6 mice 6 weeks after treatment shows a significantly lower MMP1 expression in the ITLN1 treatment group compared with the untreated controls. Bar = 10 μm. Quantification of the staining intensity for each group ($n = 10$) is shown in dot plot. $p = 0.001$; Mann–Whitney $U$ test. **g** Representative images show reduced intensity of hexose-6-phosphate (glucose-6-phosphate and fructose-6-phosphate) (ion adduct: [M-H]; mass to charge ratio (m/z): 259.0224) in tumor cells (white circles) in frozen omental tumor tissue sections from 0 to 1 h post injection of ITLN1, while increased intensity of hexose-6-phosphate is seen in adjacent adipocytes (black circles) post injection of ITLN1. The same tissue sections were stained with haematoxylin and eosin (H&E) after MALDI-imaging mass spectrometry (IMS). T tumor cells, A adipocytes; Bar = 200 μm. Three ROIs for each region (tumor and adipocyte) were randomly selected for each section ($n = 5$) based on histology of cell types for the analysis. **h** Box plot shows the normalized hexose-6-phosphate intensity in tumor cells (upper panel) and adipocytes (lower panel) at different time points after ITLN1 was intraperitoneally injected into C57BL/6 mice. ($p = 0.002$ and $0.001$, respectively; Kruskal–Wallis $H$ test). $n = 5$ in each group. **a**–**e**, **h** In the box plot, the boxes represent the interquartile range of the records, and the lines across the boxes indicate the median value of the records. The whiskers indicate the highest and lowest values among the records that are no more than 1.5 times greater than the interquartile range.

**RNA sequencing**. Total RNAs were extracted from normal mesothelial cells and cancer-associated mesothelial cells using TRI reagent (Ambion, Inc., Waltham, MA, USA). The sequencing procedures were performed as follows. In brief, RNA integrity was checked using the Agilent 2100 Bioanalyzer (Agilent Technologies, Santa Clara, CA, USA). Samples were depleted of rRNA and RNA was sheared into smaller fragments with a Covaris S220 ultrasonicator (Covaris, Woburn, MA, USA). The cDNA library was prepared using the Beckman SPRIworks system (Beckman Coulter Inc., Fullerton, CA, USA). Library fragments were hybridized to complementary oligonucleotides, and clusters of clones were generated in an Illumina cBOT instrument (Illumina, Inc., San Diego, CA, USA). Libraries were subjected to 100 cycles of paired-end sequencing (insert size, 200 base pairs) in an Illumina HiSeq 2000 system (Illumina, Inc.). Raw sequences obtained from the Illumina platform in FASTQ format were analyzed using publicly available tools. Specifically, FastQC (Babraham Institute, Babraham, UK) was used to check the quality of the raw data. After that, Tophat2 (John Hopkins University, Baltimore, MD, USA) was used for the alignment and the generation of BAM files. Up to 2 segment mismatches were allowed. Samtools (Wellcome Sanger institute, Cambridgeshire, UK) was then used to sort the BAM files by name and convert them to SAM format. Finally, HTseq (European Molecular Biology Laboratory, Heidelberg, Germany) was run in union mode to generate raw count data using the SAM files. A heat map was generated using dChip software (Affymetrix Inc.)[8].

**Quantitative RT-PCR analysis**. Total RNA was extracted from cultured cells using TRI reagent (Molecular Research Center), and 0.5 μg of total RNA was used to synthesize the first strand of cDNA using the ImProm-II Reverse Transcription System (Promega Corp., Madison, WI, USA). Pre-designed, FAM-labeled human ITLN1 (Hs00914745_m1), MMP1 (Hs00899658_m1), GLUT4 (Hs00168966_m1), and VIC-labeled human GAPDH TaqMan gene expression assays (Life Technologies Corp.) were used in the real-time PCR analysis. The relative standard curve method ($2^{-\Delta\Delta Ct}$) was used to determine the relative mRNA expression, using GAPDH as the reference.

**Western blot analysis**. Cell extracts were prepared in a RIPA buffer (20 mM sodium phosphate, 150 mM NaCl pH 7.4, 1% Nonidet P-40, 0.1% SDS, and 0.5% deoxycholic acid) containing a complete protease inhibitor cocktail (Roche Diagnostics Corp., Indianapolis, IN, USA). Proteins were separated on SDS-polyacrylamide gels and electrophoretically transferred to an Immobilon poly-vinylidene fluoride membrane (EMD Millipore, Billerica, MA, USA). The membranes were incubated with primary antibodies overnight at 4 °C, and then were incubated with appropriate horseradish peroxidase-conjugated secondary antibodies at 1:10,000 dilution (Thermo Fisher Scientific, Grand Island, NY, USA) for 1 h at ambient temperature. Signals were developed using ECL chemiluminescence detection reagents (Denville Scientific Inc., Holliston, MA, USA) and visualized on X-ray film (Fujifilm, Tokyo, Japan). The following primary antibodies were used for western blotting: anti-ITLN1 (1:1,000; AB10627; EMD Millipore), anti-MMP1 (1:1,000; AB8105; EMD Millipore), anti-p-ERK1/2 (T202/204) (1:1,000; 9101; Cell Signaling Technology, Beverly, MA, USA), anti-ERK1/2 (1:1,000; 9102; Cell Signaling Technology), anti-p-c-Jun (S73) (1:1000; 9164; Cell Signaling Technology), anti-c-Jun (1:1,000; 9165; Cell Signaling Technology), LRP1 (1:1000; LS-B2675; LifeSpan BioSciences, Inc., Seattle, WA, USA), GLUT4 (1:1000; sc-1606; Santa Cruz Biotechnology Inc., Dallas, TX, USA), and β-actin (1:5,000; clone AC-15, Sigma-Aldrich Co.). Original images for all blots are supplied as Supplementary Figs. 12–15.

**Immunohistochemical analysis**. Immunolocalization of ITLN1 and calretinin was performed using formalin-fixed, paraffin-embedded ovarian tumor sections obtained from healthy women and HGSC patients. Slides containing the sections were stained with commercially available anti-ITLN1 (1:500; A00020-01-100; Aviscera Bioscience Inc., Santa Clara, CA, USA) and anti-calretinin (1:500; NBP2-52426; Novus Biologicals, Centennial, CO, USA). Immunolocalization of LTF was performed using formalin-fixed, paraffin-embedded omental adipose tissue sections obtained from healthy women and HGSC patients. Slides containing the sections were stained with commercially available anti-LTF (1:500; PA5-80784; Thermo Fisher Scientific). Human ovarian tumor and omental adipose tissue sections from healthy women and patients with HGSC were obtained from the OC repository of the Department of Gynecologic Oncology and Reproductive Medicine under protocols approved by MD Anderson Cancer Center's institutional review board. Informed consent was obtained from all subjects. Immunolocalization of MMP1 was performed using formalin-fixed paraffin-embedded ovarian tumor sections obtained from OC-bearing C57BL/6 mice with or without ITLN1 treatment. Slides containing the sections were stained with commercially available anti-MMP1 (1:50; PA5-27210; Thermo Fisher Scientific). Target protein expression in the tumor sections from patients or mice was visualized using a Betazoid 3,3'-diaminobenzidine chromogen kit (Biocare Medical, Concord, CA, USA).

**Opal multiplex immunohistochemistry**. Multiplex immunohistochemistry was performed using sequential staining cycles as follows. In brief, formalin-fixed, paraffin-embedded omental adipose tissue sections obtained from HGSC patients were first deparaffinized and then underwent microwave treatment in citrate for antigen retrieval. They were then blocked and stained with commercially available anti-LTF (1:500, PA5-80784, Thermo Fisher Scientific), anti-CD11b (1:150, AC-0043RUO, Epitomics Inc., Burlingame, CA, USA), and anti-CD66b (1:50, NB100-77808, Novus Biologicals) at the room temperature for 1 h. Next, incubation with Opal Polymer HRP Ms + Rb (ARH1001EA, PerkinElmer Inc., Waltham, MA, USA) was performed at the room temperature for 30 min. Tyramide signal amplification (TSA) visualization was performed with Opal fluorophores. Microwave treatment was performed to remove the antibody–TSA complex after every staining cycle. Multiplex TSA was optimized by performing a triplex (CD66b, Opal 520; LTF, Opal 570 and CD11b, Opal 690) and were finished with a DAPI counterstain[33]. The tissue sections from HGSC patients were obtained from the OC repository of the Department of Gynecologic Oncology and Reproductive Medicine under protocols approved by MD Anderson Cancer Center's institutional review board. Informed consent was obtained from all subjects.

**ELISA**. Human ITLN1, LTF, TNF-α, and TGF-β concentrations in serum were measured using a commercially available ELISA kit (RD191100200R; Biovendor, Brno, Czech Republic. ORG527, BE69211, and BE69206; IBL America, Minneapolis, MN, USA, respectively) according to the manufacturer's protocol. Human serum from healthy women, patients with benign gynecologic disease and patients with HGSC were obtained from the OC repository of the Department of Gynecologic Oncology and Reproductive Medicine under protocols approved by MD Anderson Cancer Center's institutional review board. Informed consent was obtained from all subjects. Mouse ITLN1 concentrations in serum were measured using a commercially available ELISA kit (SEA933Mu; Cloud-Clone Corp., Katy, TX, USA) according to the manufacturer's protocol. Mouse blood samples were collected from C57BL/6 mice by cardiac puncture after euthanasia using a carbon dioxide chamber followed by cervical dislocation. Serum was obtained following centrifugation at 2000 $g$ for 10 min at 4 °C. The described animal procedures were reviewed and approved by the MD Anderson Cancer Center's institutional animal care and use committee.

**In vitro wound-healing assay.** Cells were seeded in 24-well plates to near confluence. Wounds were created by scratching the cell monolayer with a 200-μL pipette tip, and cultures were washed with serum-free medium to remove detached cells. Wound healing was carried out in complete medium and was photographed at 0 and 8 h after wounding. To quantify cell migration, the wound width at three different positions was measured at each time point and the migration distance was the difference in width between 0 and 8 h. The migration distance of each sample was first normalized to the initial wound width and then compared with the control sample.

**Matrigel invasion assay.** Matrigel (3.2 mg mL$^{-1}$; BD Biosciences, San Jose, CA, USA) in serum-free medium was added to each transwell polycarbonate membrane insert (8-μm pore size, 6.5-mm diameter; Corning Costar Corp., Corning, NY, USA) and allowed to dry for 1 h at 37 °C. Cells ($5 \times 10^4$) were suspended in 150 μL of serum-free medium and seeded onto the upper chambers of the transwell insert. Medium with 20% FBS (700 μL) was placed in the lower chambers. After 24 h of incubation, non-invading cells in the matrigel were removed by wiping with a cotton swab. Invading cells at the transwell membrane were fluorescently labeled with calcein (Molecular Probes, Eugene, OR, USA). Cells from five random fields were counted at ×10 magnification.

**In vitro pull-down assay.** LTF from human milk (L4894; Sigma-Aldrich Co.) and recombinant human ITLN1 (CRO104; Cell Sciences, Canton, MA, USA) were added to serum-free medium and incubated at 37 °C for 24 h. Conditioned medium was collected and incubated with anti-LTF antibody (sc-52048; Santa Cruz Biotechnology Inc.) for 2 h at 4 °C. Protein A/G PLUS-agarose beads (Santa Cruz Biotechnology Inc.) were then added to pull down bound protein–antibody complexes, and proteins were eluted from the beads with denaturing SDS buffer and subjected to western blot analysis for visualization of the ITLN1 signals.

**Duolink PLA.** Cells were cultured overnight on glass slides before fixation with 3.7% formaldehyde. Slides were then blocked with 1% bovine serum albumin and incubated with primary anti-LRP1 (1:200; ab92544; Abcam, Cambridge, MA, USA) and anti-LTF (1:100; sc-52048; Santa Cruz Biotechnology, Inc.) antibodies in 1% bovine serum albumin for 2 h at ambient temperature. A Duolink In Situ Red Starter Kit (Sigma-Aldrich Co.) was used to detect protein interaction between LRP1 and LTF according to the manufacturer's protocol. Cells were then mounted and visualized using a FluoView 1000 confocal microscope (Olympus Corp. Tokyo, Japan).

**siRNA transfection.** MMP1, LRP1, and GLUT4 were transiently silenced by siRNAs transfection (MMP1: Silencer Select siRNAs s8847 and s8848, LRP1: Silencer Select siRNAs s8278 and s8280, Thermo Fisher Scientific; GLUT4: MISSION siRNAs Hs02_00309185 and Hs02_00309186, Sigma-Aldrich Co.) duplexed with Lipofectamine RNAiMAX (Invitrogen) at a final concentration of 20, 20, and 50 nM, respectively for 24 h. Non-targeting Silencer Select siRNA and MISSION siRNA universal negative control #1 were used as negative control, respectively.

**Activated stress fiber formation.** Cells were treated with LTF with or without ITLN1 in serum-free medium for 24 h. The cells were then fixed with 3.7% formaldehyde and stained with Alexa Fluor 594 phalloidin (Life Technologies Corp.) according to the manufacturer's protocol. Fluorescent microscopy was used to evaluate F-actin rearrangement in cells.

**Intracellular calcium mobilization.** Cells were grown on collagen-coated Petri dishes with glass bottoms and loaded with Fluo-4 AM in Hank's balanced salt solution for 30 min and then followed by 20 min incubation with Hank's balanced salt solution to de-esterify the dye. Fluo-4 AM was excited at 488 nm, and its emission was collected using a bandpass filter at 535/35 nm. Fluorescent images of the cells were collected using a Leica TCS SP5 confocal microscope (Leica Microsystems, Wetzlar, Germany) at 0.25 Hz.

**Co-culture conditions.** OC cells were grown on the 0.4-μm pore size transwell insert (Thermo Fisher Scientific) for 24 h. Transwell inserts with OC cells were then put together with mesothelial cells that had been grown on the bottom well of the transwell. After 48 h, total RNA was extracted from mesothelial cells for qRT-PCR analysis to determine ITLN1 mRNA expression.

Mature adipocytes, preadipocytes, or mesothelial cells were grown on the 0.4-μm pore size transwell insert (Thermo Fisher Scientific) and treated with ITLN1 and/or LTF for 24 h. Transwell inserts with cells were then put together with OC cells that had been grown on the bottom well of the transwell. After 72 h, MTT assay and lactate secretion assay were performed to determine cell viability and lactate secretion, respectively.

**MTT assay.** Cells were incubated with 1 mg mL$^{-1}$ 3-(4,5-dimethylthiazol-2-yl)-2,5-diphenyltetrazolium bromide (MTT) (Sigma-Aldrich Co.) in PBS for 3 h. The formazan that formed was then solubilized by adding dimethyl sulfoxide. The absorbance was read at 570 nm using a FLUOstar Galaxy plate reader (BMG Labtech, Offenburg, Germany).

**Glucose consumption assay.** The assay was performed using Wako Glucose Kit (Wako Diagnostics, Mountain View, CA, USA) according to the manufacturer's protocol. In brief, 250 μL of reconstituted Wako glucose reagent was added to a 2 μL sample in a 96-well plate and incubated with shaking at 37 °C for 5 min. The absorbance was measured at 505 nm using a SpectraMax M5 spectrophotometer (Molecular Devices, Sunnyvale, CA, USA). The change in absorbance between samples with or without treatment indicated the glucose uptake of cells.

**Lactate secretion assay.** The assay was performed using Trinity Lactate Kit (Trinity Biotech Plc., Co Wicklow, Ireland) according to the manufacturer's protocol. In brief, reconstituted lactate reagent was added to 1:10 diluted media samples in an assay plate. The plate was incubated at 37 °C for 1 h and protected from light, and the absorbance was read at 540 nm using a spectrophotometer. The change in absorbance between samples with or without ITLN1 treatment indicated the lactate secretion of cells.

**Isotope tracer analysis using GC-MS.** Cells co-cultured with ITLN1-treated mature adipocytes were incubated with medium containing U-$^{13}$C$_6$ glucose for 24 h before metabolite extraction. The extraction and subsequent GC–MS procedures were performed as follows. In brief, cells were washed with ice-cold PBS once and quenched with 400 μL ice-cold methanol. Then, 400 μL of water containing 1 μg norvaline (Sigma-Aldrich Co.) was added. Cells were then scraped into microfuge tubes and vortexed at 4 °C with 800 μL chloroform for 30 min. Next, the mixture was centrifuged at 5000 g at 4 °C for 10 min. The aqueous layer was collected for metabolite analysis. Aqueous samples were dried and dissolved in 30 μL of 2% methoxyamine hydrochloride in pyridine (Pierce Biotechnology, Inc., Waltham, MA, USA) and sonicated for 10 min. Afterwards, samples were kept at 37 °C for 2 h. Then, after addition of 45 μL MTBSTFA with 1% TBDMCS (Pierce Biotechnology, Inc.) for derivatization, samples were kept at 55 °C for 1 h. GC–MS analysis was performed using an Agilent 6890 GC equipped with a 30-m Rtx-5 capillary column connected to an Agilent 5975B MS (Agilent Technologies). The following thermal gradient was used for the GC oven: 100 °C for 3 min, followed by a temperature increase of 5 °C/min to 300 °C for a total run time of 48 min. Data were acquired in scan mode[34]. The abundance of relative metabolites was calculated from the integrated signal of all potentially labeled ions for each metabolite fragment. Mass isotopologue distributions were corrected for natural abundance using IsoCor software (LISBP, Toulouse, France) prior to analysis.

**In vivo studies.** Female C57BL/6 mice at the age of 6 weeks old were used. Mice were maintained in barrier animal facilities approved by the American Association for Accreditation of Laboratory Animal Care (AAALAC) and in accordance with current regulations and standards of the United States Department of Agriculture (USDA), Department of Health and Human Services (DHHS), and National Institutes of Health (NIH). All Department of Veterinary Medicine and Surgery facilities' environmental conditions are monitored electronically by both Facilities Management and Edstrom Watchdog environmental monitoring system. Each room has designated temperature (ambient temperature) and humidity (40–60%) set-points and acceptable ranges. Lighting conditions (on and off) are also monitored. Animal room lights are controlled to turn on and off at designated times (12-h light/dark cycle).

To determine the effect of OC cells on serum ITLN1 levels in vivo, mouse OC IG10 cells ($2 \times 10^6$) were injected intraperitoneally into female C57BL/6 mice at the age of 6 weeks old. Mice were euthanized using a carbon dioxide chamber followed by cervical dislocation after 6 weeks. Blood samples were collected, and serum ITLN1 levels were measured using a commercially available ELISA kit (Cloud-Clone Corp.) as described above.

To determine the duration of effective action of recombinant mouse ITLN1 in vivo, luciferase-labeled mouse OC IG10 cells ($2 \times 10^6$) were injected intraperitoneally into female C57BL/6 mice at the age of 6 weeks old to establish tumors. After 7 days, mice were randomized and treated with one dose of recombinant mouse ITLN1 (2 mg kg$^{-1}$; GenScript, Piscataway, NJ, USA). Blood samples were collected at 0, 1, 6, 12, 24, and 48 h to measure glucose and serum ITLN1 levels using the AlphaTRAK 2 blood glucose monitoring system (Abbott Laboratories, Chicago, IL, USA) and a commercially available ELISA kit (Cloud-Clone Corp.) as described above, respectively. Omental tissues with tumors were snap frozen in liquid nitrogen and processed for MALDI-IMS.

To determine the effect of recombinant mouse ITLN1 on OC growth in vivo, luciferase-labeled mouse OC IG10 cells ($2 \times 10^6$) were injected intraperitoneally into female C57BL/6 mice at the age of 6 weeks old to establish tumors. After 7 days, mice were randomized and treated with 2 mg kg$^{-1}$ recombinant mouse ITLN1 every 2 days for 6 weeks. The tumor volumes were measured and quantified using the IVIS-Lumina XR in vivo imaging system (Caliper Life Sciences, Inc., Hopkinton, MA, USA). Mice were euthanized using a carbon dioxide chamber followed by cervical dislocation. Blood samples were collected to measure glucose and serum ITLN1 levels as described above. Ascites volumes were measured, and

omental tissues with tumors were fixed in formalin and processed for immunohistochemistry.

The described animal procedures were reviewed and approved by MD Anderson Cancer Center's institutional animal care and use committee.

**MALDI-IMS**. Sections from fresh-frozen mouse omental tumor tissues were obtained at 15 μm thickness in a cryostat and mounted on glass microscope slides. HTX Sprayer M5 (HTX Technologies, LLC., Chapel Hill, NC, USA), an automated sprayer matrix applicator was used to coat the tissue sections with 10 mg mL$^{-1}$ 1,5-diaminonaphthalene (Sigma-Aldrich Co.) as the MALDI matrix dissolved in 90/10 acetonitrile/water. The slides were scanned using an EPSON scanner (Epson, Suwa, Japan), and the areas of interest were mapped into High Definition Imaging software (HDI 1.4; Waters, Milford, MA, USA) before being loaded into the mass spectrometer.

Imaging of tissue sections was achieved with a SYNAPT G2-Si high resolution mass spectrometer with a MALDI source interface (Waters), specifically configured for imaging biological and chemical materials. The scanner used a 2.5 kHz Nd:YAG solid state laser that rastered across the tissue sample, giving a chemical composition profile for each corresponding spatial coordinate. These mass spectral data were collated by HDI to produce a chemical image that could be correlated with the sample histological profile. The laser power was set to 250 (arbitrary units) with 300 laser shots per pixel of data. The laser raster step was set to 60 μm to match the laser spot size of 60 μm. The instrument was checked for mass accuracy using red phosphorus, and mass calibrated with the m/z range from 50 to 1000 before acquiring the data. After negative mode acquisition, HDI automatically processed the raw data into a collection of images. Heat map images that is correlated to a compound of interest can be extracted.

The tissue sections were washed and stained with hematoxylin and eosin (H&E) after acquisition. By overlaying the images of H&E staining and heat map images using HDI according to the shape of the tissue section, three regions of interest (ROIs) were identified and selected randomly according to the histology of cell types (tumor and adipocytes). The ROI raw data were then imported into data analysis software Progenesis QI (Waters) for metabolite identification. The search criteria are as follows, (i) mass resolution is 20,000; (ii) pixel size is 60 × 60 μm; (iii) ion adduct formed is [M-H]$^-$ for negative mode; and (iv) the database for metabolite annotation is Human Metabolome Database.

Normalization of metabolite abundance was done in data analysis software Progenesis QI using total ion count (TIC). In summary, Progenesis QI uses a median and mean absolute deviation approach based on all the detected abundance to calculate quantitative abundance ratios that models the differences in ion abundances between samples and apply an approximation of the mean and variance of the ratio distribution to finalize the calculation of the scaling factor. Further details are found in http://www.nonlinear.com/progenesis/qi/v1.0/faq/how-normalisation-works.aspx (Progenesis QI).

**Statistical analysis**. SPSS software, version 23 (IBM Corp. Armonk, NY, USA), and R (3.4.0; The R Foundation) were used to perform statistical tests. Data are presented as mean ± the standard deviation unless otherwise specified. A two-tailed Student's $t$ test was used to test differences in sample means for data with normally distributed means. The Mann–Whitney $U$ test or Kruskal–Wallis $H$ test was used for nonparametric data as appropriate. Correlation between variables was determined using a Spearman's correlation analysis. ROC test function with the default method was used for comparison between areas under curves[35]. A $p$ value of <0.05 was considered to be statistically significant.

**Reporting summary**. Further information on research design is available in the Nature Research Reporting Summary linked to this article.

## Data availability
The transcriptome profiling data for microdissected normal and cancer-associated adipose tissue have been deposited in the Gene Expression Omnibus Data Bank (GEO; National Cancer for Biotechnology Information, Bethesda, MD, USA) under the accession code GSE120196 [https://www.ncbi.nlm.nih.gov/geo/query/acc.cgi?acc=GSE120196]. The RNA-sequencing data for normal and cancer-associated mesothelial cells have been deposited in GEO under the accession code GSE120303 [https://www.ncbi.nlm.nih.gov/geo/query/acc.cgi?acc=GSE120303]. The transcriptome profiling data for ovarian cancer A224 cells with ITLN1 treatment have been deposited in GEO under the accession code GSE120245 [https://www.ncbi.nlm.nih.gov/geo/query/acc.cgi?acc=GSE120245]. All the other data supporting the findings of this study are available within the article and its supplementary information files and from the corresponding author upon reasonable request. A reporting summary for this article is available as a Supplementary Information file.

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

## Acknowledgements

This study was supported in part by grants R01CA184918, R01CA227622, U01188388, The University of Texas MD Anderson Cancer Center Support Grant P30CA016672 from the National Institutes of Health; Wiegand Foundation and James B. & Lois R. Archer Charitable Foundation through the Center for Radiation Oncology Research; grants W81XWH-17-0126 and W81XWH-17-1-0146 from the Ovarian Cancer Research Program, Department of Defense; grant 318159 from the American Institute for Cancer Research; and T.T. & W.F. Chao Foundation and John S. Dunn Research Foundation. We thank MD Anderson's Department of Scientific Publications for providing editorial assistance.

## Author contributions

C.-L.A.-Y. and S.C.M. planned the experiments. C.-L.A.-Y. conducted the majority of the experiments and analyzed data. C.-L.A.-Y., T.-L.Y. and N.-N.C. performed microarray and RNA-sequencing experiments and analyzed the data. C.-L.A.-Y. and S.-Y.K. performed in vivo experiments. A.A. and H.Z. performed the GC–MS experiment. K.-P.Y. performed calcium measurement experiment. M.O. performed ELISA on clinical samples. C.-L.A.-Y., J.S., Y.Z., and S.T.C.W. analyzed clinical data. D.L.B. performed MALDI-IMS experiment. M.L.A. provided omental tissue samples. A.R.-V., R.S., K.H.L., S.T.C. W., D.N. provided intellectual contributions to experimental design. C.-L.A.-Y., D.N., and S.C.M. prepared and revised the manuscript.

## Competing interests

The authors declare no competing interests.
