## [Peer Review File · Nature Communications]

Reviewers' comments:

Reviewer #1 (Remarks to the Author):

Summary: In this manuscript, the authors showed that mesothelial cells in the omental microenvironment modulate the invasive potential and proliferation of ovarian cancer (OC) cells via ITLN1. The expression level of ITLN1 was down-regulated in mesothelial cells enriched in the OC omental adipose tissue compared to normal controls. The authors showed that pro-inflammatory cytokines released from the OC-associated omental microenvironment decrease the expression level of ITLN1 in mesothelial cells. They also demonstrated that ITLN1 suppresses the motility and invasive potential of OC cells via the ITLN1-LTF-MMP1 axis. In addition, ITLN1 suppresses the proliferation of OC cells by increasing glucose uptake in adipocytes through upregulation of GLUT4, which was supported by both in vitro and in vivo observations.

The present manuscript makes a valuable contribution by providing a mechanistic insight into the cross-talk between mesothelial cells in the OC-associated omental microenvironment and OC cells. Furthermore, they demonstrated the feasibility of using ITLN1 as a therapeutic agent for OC treatment in vivo. However, their proposed mechanism regarding the ITLN1-LTF axis requires further in vivo validation.

Major points:

1. Supplementary Figure 1a: The authors showed that mesothelial cells are the major source of ITLN1 by comparing its expression levels among diverse cell types enriched in the omental adipose tissue. However, they used cell lines whose genetic background and culture conditions are not identical, which possibly affects the expression levels of genes. A more direct comparison using cells isolated from the omental adipose tissue (e.g. single-cell RNA-seq) should be applied.

2. Supplementary Figure 1b-c: The in vitro data suggest that pro-inflammatory cytokines such as TNF-alpha and TGF-beta downregulate the expression of ITLN1 in mesothelial cells of OC patients. The cross-talk between immune cells and mesothelial cells via cytokines is one of the key findings in this manuscript, but this was not strongly supported by in vivo data. I'm wondering whether these pro-inflammatory cytokines are more highly expressed in OC patients than in normal controls (using serum or the omental adipose tissue). What are the major cell types secreting these cytokines in the omental microenvironment? Are the corresponding cytokine receptors upregulated in mesothelial cells of OC patients? The repertoires of cytokines affecting the expression level of ITLN1 in mesothelial cells should be systematically examined by analyzing the transcriptome data.

3. Figure 2e: I'm wondering whether the AUC score was calculated using cross-validation. I could not find the details of the logistic regression analysis. Without cross-validation, the prediction model with more features are more prone to over-fitting.

4. LTF and MMP1: Which cell types in the omental microenvironment are the major source of LTF? The authors mentioned that neutrophils might be a candidate, but this should be validated. Are the levels of LTF up-regulated in the omental adipose tissue or peritoneal fluid of OC patients compared to normal controls? Is there any correlation between circulating LTF and ITLN1 in OC patients? I'm also wondering whether we can use LTF as a diagnostic marker in combination with CA125 and ITLN1.

5. Figure 6: ITLN1 seems to increase glucose uptake in adipocytes by upregulating GLUT4 in a LTF-dependent manner. I'm wondering whether ITLN1 can upregulate GLUT4 in adipocytes cultured in SFM without LTF.

Minor points:

1. Figure 1a: ITNL1 should be indicated.

Reviewer #2 (Remarks to the Author):

This is a very comprehensive study that includes observational data from human tissues, blood and cell lines, functional and biochemical data and in vivo studies to show the cellular interplay in the tumor microenvironment of mesothelial cells and adipocytes and their impact on ovarian cancer cells. Specifically, the authors show that circulating ITLN1 is low in women with high grade ovarian cancer, that high ITLN1 levels decrease ovarian cancer cell proliferation, motility and invasion and that ITLN1 is produced only in mesothelial cells. They go on to show that ITLN1 increases glucose uptake in mature adipocytes and this leads to glucose starvation of ovarian cancer cells and the resulting decreased invasive potential.

The study is novel and will be of great interest to the ovarian cancer and general cancer research community. My concerns are related to the lack of detail in the Methods throughout the manuscript that made it difficult to interpret the data. Some examples of concerns are detailed below.

In Results it states that mesothelial cells were isolated from the peritoneal fluid of healthy women. There is no reference to this in the methods and it is unclear how this is possible. Please clarify and correct.

The manuscript includes the use of tissues from women. However, no ethics information is supplied. Please add in the appropriate ethics protocol identifiers.

Figure 1e – tissues were from non-ovarian cancerous and ovarian cancerous tissues, but not normal tissue as stated. Please correct here and throughout.

Confirmation of preadipocytes and mature adipocytes needs to be shown.

There is no information on how the co-culture experiments (Fig 1f,g) were performed. Please clarify.

No method information for Sup Figs 1b, c.

Methods for siRNA transfections and effect on target genes / proteins (MMP1 and LRP1) need to be included

Cell treatments and conditions for Fig 5 need to be included. Are the values shown the result of 3 independent experiments? Statistical data showing significance needs to be included.

Conditions and timings for the MTT assays need to be included (SI Fig 4)

Co-culture conditions (Fig 4 and SI fig 6) need to be included.

They report from the literature but do not demonstrate that ITLN1 increases insulin-dependent glucose uptake exclusively in adipocytes. They show that ITLN1 in the presence of mature adipocytes decreased the growth of ovarian cancer cells. However, the methods state that the only cells that were grown with insulin were the adipocytes, thus these were essentially the only cells tested under the appropriate conditions. Their conclusion that only mature adipocytes play a role in mediating ITLN1's growth suppressive effects on cancer cells requires that all cells be grown in the same conditions. Similarly, GLUT4 is regulated by insulin, so different results would be expected in the cells grown with and without insulin.

Reviewer #3 (Remarks to the Author):

The manuscript by Au-Yeng and colleagues provides evidence that the expression of intelectin-1 (ITLN1), an intestinal lactoferrin (LTF) receptor, is down-regulated in ovarian cancer (OC) associated mesothelial cells. Decreased ITLN1 expression is found in serum samples of OC patients and in mice bearing OC. Serum ITLN1 levels have prognostic significance in OC patients. Mechanistically, they show that ITLN1 represses OC migration/invasion and the expression of collagenase (MMP1).

Apparently, the ITLN1-LTF interaction prevents LTF to interact with its receptor LRP1, which is responsible for the ERK1/2 mediated activation and upregulation of MMP1 expression, thus resulting in the attenuation of OC invasive potential. Co-culture experiments of OC cells with adipocytes in the presence of ITLN1 indicated a growth-suppressive effect of ITLN1 on OC cells that was abrogated by the addition of glucose. These findings suggested that the ITLN1-induced glucose uptake in adipocytes restricted glucose utilization and limited growth of OC cells. Consistently, they show that ITLN1 upregulates GLUT4 expression in the adipocytes and show that GLUT4 is required to abrogate OC cells growth by ITLN1. Interestingly, LTF downregulates GLUT4 expression and glucose uptake leading to enhanced OC growth. Analysis of the glycolytic flux by determination of the lactate released in the medium and by the production of labeled pyruvate and lactate by GC-MS, indicated that ITLN1 diminished the glycolytic flux of OC cells co-cultured with adipocytes, suggesting that ITLN1-treated adipocytes inhibit the glycolytic of OC cells. Finally, they show that ITLN1 administration to mice bearing OC suppresses MMP1 expression and arrests tumor growth in vivo. By using a cutting edge MALDI-IMS approach to analyze metabolites in tissue sections from the omental tumors derived in mice, they report a rapid 1h reduction of glucose-6-phosphate and lactate content in tumor areas whereas the adjacent adipocytes showed the opposite trend in both metabolites in response to ITLN1 administration. Overall, the manuscript supports both in vitro and in vivo that ITLN1 suppresses tumor growth by limiting the glucose available to OC cells by an unfavorable competition with the nearby adipocytes.

I find this paper solid, addressing mechanistic aspects and very convincing. The MALDI-imaging mass spectrometry approach nicely documents a rapid opposite effect of ITLN1 administration in glucose utilization and glycolytic flux in adipocytes when compared to the nearby tumor cells.

Response to Referees # NCOMMS-19-01545A

Reviewer #1

Comment: Supplementary Figure 1a: The authors showed that mesothelial cells are the major source of ITLN1 by comparing its expression levels among diverse cell types enriched in the omental adipose tissue. However, they used cell lines whose genetic background and culture conditions are not identical, which possibly affects the expression levels of genes. A more direct comparison using cells isolated from the omental adipose tissue (e.g. single-cell RNA-seq) should be applied.

Response: In order to demonstrate that mesothelial cells are the major source of ITLN1 and the spatial distribution of ITLN1 expressing cells, we performed immunolocalization of ITLN1 and calretinin (a known mesothelial cell marker) on formalin-fixed paraffin-embedded (FFPE) omental tissue sections from healthy women and patients with HGSC instead of using single-cell RNA-seq. We found that ITLN1 is highly expressed in normal adipose tissues but not in cancer-associated adipose tissues. The expression of ITLN1 is also highly co-localized with calretinin positive mesothelial cells covering the omental adipose tissue but not in other cell types. Representative microscopic images are presented as Fig. 1e and are described in the Results section on page 4 of the revised manuscript.

Comment: Supplementary Figure 1b-c: The in vivo data suggest that pro-inflammatory cytokines such as TNF-alpha and TGF-beta downregulate the expression of ITLN1 in mesothelial cells of OC patients. The cross-talk between immune cells and mesothelial cells via cytokines is one of the key findings in this manuscript, but this was not strongly supported by in vivo data. I'm wondering whether these pro-inflammatory cytokines are more highly expressed in OC patients than in normal controls (using serum or the omental adipose tissue).

Response: The expression levels of TNF- α and TGF- β in serum samples from healthy women and HGSC patients are measured using commercially available ELISA kit (BE69211 and BE69206, respectively; IBL America) according to the manufacturer's protocol. The mean levels of both TNF- α and TGF- β are higher in serum from HGSC patients compared to that from healthy women (TNF- α mean level: 116 vs 104 pg/mL, TGF- β mean level: 252 vs 154 pg/mL). However, the difference does not reach significance. The results are presented in Supplementary Fig. 1d and 1e, and are described in the Results section on page 5 of the revised manuscript.

Comment: What are the major cell types secreting these cytokines in the omental microenvironment?

Response: Cytokines like TNF- α and TGF- β are secreted by many cell types. The major cell type that produce TNF- α is macrophages. Besides, natural killer cells, neutrophils, mast cells, endothelial cells, adipose tissues and fibroblasts that are present in the omental microenvironment also produce TNF- α . (Parameswaran and Patial, 2010) For TGF- β , it is also mainly expressed in the immune system including macrophages. (Wrzesinski et al., 2007, Yang et al., 2013)

Comment: Are the corresponding cytokine receptors upregulated in mesothelial cells of OC patients? The repertoires of cytokines affecting the expression level of ITLN1 in mesothelial cells should be systematically examined by analyzing the transcriptome data.

Response: The RNA-seq data on ovarian cancer-associated mesothelial cells and normal mesothelial cells showed that TNF-alpha (TNFRSF1A and TNFRSF1B) and TGF-beta (TGFB2 and TGFB3) receptors are upregulated in mesothelial cells from OC patients. Meanwhile, TNF-alpha and TGF-beta were also shown to be upregulated in cancer-associated mesothelial cells compared to normal mesothelial cells. The data are summarized as Supplementary Table 2 in the revised manuscript. Although the circulating levels of TNF-alpha and TGF-beta are not significantly higher in HGSC patients than in healthy women, the localized levels of these cytokines in mesothelial cells are increased, so do the levels of their corresponding receptors. This suggests that the expression level of ITLN1 in mesothelial cells in HGSC patients is regulated by these cytokines in an autocrine and paracrine manner in the tumor microenvironment. The results are described in the Results section on page 5 and Discussion section on pages 14 and 15 of the revised manuscript.

Comment: Figure 2e: I'm wondering whether the AUC score was calculated using cross-validation. I could not find the details of the logistic regression analysis. Without cross-validation, the prediction model with more features are more prone to over-fitting.

Response: The cross-validation is a good technique to optimize hyperparameters and reduce the chance of overfitting for a model with high dimensional feature space. However, in our case, we only tested the diagnostic value of one or two features, i.e., the ROC curves were generated based on the gene expression value of one feature (ITLN1) or two features (ITLN1 and CA125). Also, we did not include any hyperparameters in our model because it was not necessary to constrain the weights of the one or two given features. Cross-validation will be used for calculation if we want to select multiple markers from feature space containing hundreds or thousands of genes in the feature.

Comment: LTF and MMP1: Which cell types in the omental microenvironment are the major source of LTF? The authors mentioned that neutrophils might be a candidate, but this should be validated.

Response: LTF has been reported to be mainly found in human neutrophils. We have this observation validated in omental tumor tissues from HGSC patients using Opal multiplex immunohistochemistry. Cancer-associated omental tissues were stained with CD11b, CD66b and LTF together with DAPI (nuclear stain). CD11b and CD66b are used as neutrophil markers. We found that most of the LTF present in the tissues are co-localized with CD11b and CD66b. This confirms that neutrophil is the major source of LTF. Representative images are presented in Supplementary Fig. 2a-2f and are described in the Results section on page 7 of the revised manuscript.

Comment: Are the levels of LTF up-regulated in the omental adipose tissue or peritoneal fluid of OC patients compared to normal controls?

Response: The expression levels of LTF were examined in cancer-associated adipose tissues from HGSC patients and normal adipose tissues from healthy women using immunohistochemical analysis. The results showed a significant upregulation of LTF in ovarian cancer-associated adipose tissue compared to normal adipose tissues (n=7 for each group, p=0.001). The results are presented in Supplementary Fig. 2g and are described in the Results section on page 7 of the revised manuscript.

In addition, the level of LTF in sera from healthy women, patients with benign diseases and

HGSC patients, and that in ascites from HGSC patients are measured using a commercially available ELISA kit (ORG 527; IBL America) according to the manufacturer's protocol. We found that there is a trend of increasing levels of LTF in sera from healthy women to HGSC patients although the change does not reach significance. However, the level of LTF in ascites, which is rich in neutrophils, from HGSC patients is significantly higher than that in sera from any of the group examined. The results are presented in Supplementary Fig. 2h and are described in the Results section on page 8 of the revised manuscript.

Comment: Is there any correlation between circulating LTF and ITLN1 in OC patients?

Response: Human LTF concentrations in sera were measured using a commercially available ELISA kit (ORG 527; IBL America) according to the manufacturer's protocol. Human sera from healthy women, patients with benign gynecologic disease and patients with HGSC were obtained from the ovarian cancer repository of the Department of Gynecologic Oncology and Reproductive Medicine under protocols approved by MD Anderson Cancer's institutional review board. The ELISA results showed that there were no significant correlation between circulating LTF and ITLN1 levels in OC patients ($r=0.021$; $p=0.738$). The results are presented in Supplementary Fig. 3a and are described in the Results section on page 8 of the revised manuscript.

Comment: I'm also wondering whether we can use LTF as a diagnostic marker in combination with CA125 and ITLN1.

Response: Receiver operating characteristic curves were constructed for LTF, LTF in combination with CA125, and with CA125 and ITLN1 to test for discriminatory ability between healthy women and women with OC. We found that LTF alone had a significantly smaller AUC than CA125 alone ($p=2.2e-16$), and that for LTF with CA125 does not have a significant larger AUC than CA125 alone ($p=0.448$) (Supplementary Fig. 3c). However, ROC curve for LTF in combination with ITLN1 and CA125 showed a significant larger AUC than CA125 alone ($p=0.005$) or CA125 with ITLN1 ($p=0.037$) (Fig. 2e). These data suggest that LTF in combination with ITLN1 complements CA125 in identification of OC patients. The ROC curves for LTF and LTF with CA125 are presented in Supplementary Fig. 3c while that for LTF in combination with ITLN1 and CA125 are presented in Fig. 2e, and are described in the Results section on page 8 of the revised manuscript.

Comment: Figure 6: ITLN1 seems to increase glucose uptake in adipocytes by upregulating GLUT4 in a LTF-dependent manner. I'm wondering whether ITLN1 can upregulate GLUT4 in adipocytes cultured in SFM without LTF.

Response: Quantitative RT-PCR analysis was used to determine the expression of GLUT4 mRNA in mature adipocytes treated with ITLN1 in SFM. The results showed that GLUT4 mRNA expression was not significantly upregulated in mature adipocytes treated with ITLN1 in SFM compared to controls. The results are presented in Supplementary Fig. 7d and are described in the Results section on page 11 of the revised manuscript.

Comment: Figure 1a: ITLN1 should be indicated.

Response: Figure 1a is revised according to the reviewer's comment with ITLN1 indicated.

Reviewer #2

Comment: In Results it states that mesothelial cells were isolated from the peritoneal fluid of healthy women. There is no reference to this in the methods and it is unclear how this is possible. Please clarify and correct.

Response: The normal mesothelial cells were isolated from omental adipose tissue from consented non-oncological adult donors undergoing elective gastric bypass surgery (Zen-Bio, Inc., Research Triangle Park, NC, USA) or from peritoneal washing from patients with benign gynecologic diseases. The Results section on page 4 is amended accordingly and a reference for the method of isolating mesothelial cells from peritoneal washing is included in the Methods section under the “Cell lines and culture conditions” subsection in the revised manuscript.

Comment: The manuscript includes the use of tissues from women. However, no ethics information is supplied. Please add in the appropriate ethics protocol identifiers.

Response: All human tissues were obtained from the ovarian cancer repository of the Department of Gynecologic Oncology and Reproductive Medicine under protocols approved by MD Anderson Cancer Center’s institutional review board. Informed consent was also obtained from all subjects. This declaration is included in the Methods section under the “Immunohistochemical analysis” subsection in the revised manuscript.

Comment: Figure 1e – tissues were from non-ovarian cancerous and ovarian cancerous tissues, but not normal tissue as stated. Please correct here and throughout.

Response: The omental adipose tissue sections shown in Fig. 1e were from healthy women and HGSC patients.

Comment: Confirmation of preadipocytes and mature adipocytes needs to be shown.

Response: Mature adipocytes (mADSC) were differentiated from adipose-derived stem cells (ADSC; preadipocytes) purchased from ATCC under three cycles of differentiation. Detailed differentiation conditions are provided in the Methods section under the “Cell lines and culture conditions” subsection in the revised manuscript. The cells were characterized using Oil Red O staining. Representative images are presented in Supplementary Fig. 6b and are described in the Results section on page 10 of the revised manuscript.

Comment: There is no information on how the co-culture experiments (Fig 1f, g) were performed. Please clarify.

Response: The method of co-culture experiments in Figure 1f and g are as follows, Ovarian cancer cells were grown on the 0.4 μ m pore size transwell insert (Thermo Fisher Scientific) for 24 h. Transwell inserts with ovarian cancer cells were then put together with mesothelial cells that had been grown on the bottom well of the transwell. After 48 h, total RNA was extracted from mesothelial cells to determine the ITLN1 mRNA expression using qRT-PCR analysis.

It is also described in the Methods section under the “Co-culture conditions” subsection in the revised manuscript.

Comment: No method information for SI Figs 1b, c.

Response: The method and conditions for Supplementary Fig. 1b and c are included in the

figure legends of Supplementary Fig. 1 in the revised manuscript.

Comment: Methods for siRNA transfections and effect on target genes / proteins (MMP1 and LRP1) need to be included.

Response: The method for siRNA transfections are as follows, MMP1, LRP1 and GLUT4 were transiently silenced by siRNAs transfection (MMP1: Silencer Select siRNAs s8847 and s8848, LRP1: Silencer Select siRNAs s8278 and s8280, Thermo Fisher Scientific; GLUT4: MISSION siRNAs Hs02_00309185 and Hs02_00309186, Sigma-Aldrich Co.) duplexed with Lipofectamine RNAiMAX (Invitrogen) at a final concentration of 20, 20 and 50 nM, respectively for 24 h. Non-targeting Silencer Select siRNA and MISSION siRNA universal negative control #1 were used as negative control respectively.

It is also described in the Methods section under the “siRNA transfection” subsection in the revised manuscript.

The effect of MMP1 and LRP1 siRNAs transfection on MMP1 and LRP1 genes and proteins are presented in Supplementary Fig. 4c-d (MMP1) and Supplementary Fig. 5a-b (LRP1) and are described in the Results section on pages 8 and 9 of the revised manuscript.

Comment: Cell treatments and conditions for Fig 5 need to be included. Are the values shown the result of 3 independent experiments? Statistical data showing significance needs to be included.

Response: The cell treatments and conditions for Fig. 5 are as follows,

Fig. 5a: SKOV3 and A224 cells were incubated in SFM with or without 100 µg/mL LTF for 24 h.

Fig. 5b: SKOV3 and A224 cells were treated with 100 µg/mL LTF in SFM with or without 500 ng/mL ITLN1 for 24h.

Fig. 5c: SKOV3 and A224 cells were transfected with LRP1-specific siRNAs or negative control siRNA for 24 h before 24-hour treatment of LTF in SFM.

The related Methods and Figure Legend sections were revised accordingly in the revised manuscript.

The values presented in Fig. 5 are relative normalized protein levels with respect to the corresponding control for the Western blots shown. The experiments were repeated three times independently. The average values of the normalized protein levels and statistical analyses for the three experiments are presented in Supplementary Fig. 10 of the revised manuscript.

Comment: Conditions and timings for the MTT assays need to be included (SI Fig 4)

Response: Conditions and timings for the MTT assays in Supplementary Fig. 4 (a, b, d-f, i) (now Supplementary Fig. 6a, 6c-6f and 7e in the revised manuscript) are included in the figure legends of Supplementary Fig. 6 and 7 in the revised manuscript.

Comment: Co-culture conditions (Fig 4 and SI Fig 6) need to be included.

Response: The method of co-culture experiments in Figure 6 and SI Fig 4 are as follows,

Mature adipocytes, preadipocytes or mesothelial cells were grown on the 0.4 µm pore size transwell insert (Thermo Fisher Scientific) and treated with ITLN1 and/or LTF for 24 h. Transwell inserts with cells were then put together with ovarian cancer cells that had been grown on the bottom well of the transwell. After 72 h, MTT assay and lactate secretion assay were performed to determine cell viability and lactate secretion, respectively.

The co-culture experimental protocol is also described in the Methods section under the “Co-culture conditions” subsection in the revised manuscript.

Comment: They report from the literature but do not demonstrate that ITLN1 increases insulin-dependent glucose uptake exclusively in adipocytes. They show that ITLN1 in the presence of mature adipocytes decreased the growth of ovarian cancer cells. However, the methods state that the only cells that were with insulin were the adipocytes, thus there were essentially the only cells tested under the appropriate conditions. Their conclusion that only mature adipocytes play a role in mediating ITLN1's growth suppressive effects on cancer cells requires that all cells be grown in the same conditions. Similarly, GLUT4 is regulated by insulin, so different results would be expected in the cell grown with and without insulin.

Response: We also include the effect of ITLN1 on glucose uptake in both mature adipocytes and preadipocytes in the absence of insulin in the revised Fig. 6b. The results show that ITLN1 has no significant effect on glucose uptake in both mature adipocytes and preadipocytes in the absence of insulin while significant increase in glucose uptake was only observed in mature adipocytes in the presence of insulin but not in preadipocytes. These results demonstrate that ITLN1 increases insulin-dependent glucose uptake exclusively in mature adipocytes. Based on this finding, all the other experiments using ITLN1-treated mature adipocytes to demonstrate the effect of glucose uptake on ovarian cancer cell growth, GLUT4 mRNA and protein expressions, lactate secretion and GC-MS are performed in the presence of insulin unless otherwise specified. The related figure legend was revised to address reviewer's concern in the revised manuscript. The effect of ITLN1 on GLUT4 mRNA expression in mature adipocytes was also examined in the absence of insulin. We found that there is no significant increase in GLUT4 mRNA expression after ITLN1 treatment in the absence of insulin. The results are presented in Supplementary Fig. 7a and are described in the Results section on page 11 of the revised manuscript.

Reviewers' comments:

Reviewer #1 (Remarks to the Author):

My concern has been addressed by the authors.

Reviewer #2 (Remarks to the Author):

The authors have addressed the original comments.

A few points remain:

The western blot results appear to be from single, non-replicated experiments (eg, Figs 3e, 5, SI5c) rather than representative of replicate independent experiments. If this is the case, the results are not sufficient for the mechanistic interpretations presented from these results. Please clarify / add experiments and statistical tests.

Many journals now also require the full blots in SI. Please check if this is required.

LTF levels are graphed in SI Fig 2h and SI3b as U/mL. In line 196 it states that ovarian cancer cells with treated" ... with 100 ug/ml (a physiological level) of LTF in serum." I cannot find the conversion from Units to ug. Please include this in either the results or methods.

Mouse experiments:

Methods line 660-661 – “ ... IG10 cells were injected intraperitoneally into female C57BL/6 mice for 6 weeks” Please correct – cells should only be injected once.

Figure 7. Please include the timepoint shown for each experiment.

Was there an increase in survival for the mice treated with recombinant ITLN1?

Reviewer #4 (Remarks to the Author):

This reviewer has focused primarily on the MALDI MSI data part of the manuscript, as this is my area of expertise. There are a number of questions concerning the MALDI MSI data which should be addressed (but I am confident that they are addressable) before the manuscript may be considered for publication.

MALDI Imaging data

- i) Figure 7H and Supplementary Figures 8f and 8g. The figures claim to report the normalized abundances of glucose-6-phosphate, lactate and ATP. It is not explained how the data was normalized, as such it is not possible to understand what metric is being displayed in the MSI figures.
- ii) Supplementary Figures 8f. The figure caption states “lactate abundance (ion adduct: [M-H]⁻; m/z: 88.01604).” However the m/z is not correct. In atomic mass units the correct m/z of the [M-H]⁻ anion is 89.0244176 (neutral molecule is 90.03169406). It is not clear how the authors have assigned an incorrect mass to lactate.
- iii) Supplementary Figures 8f. The m/z given for the ATP [M-H]⁻ anion is not correct. The mass they have given is that of the [M-H] neutral molecule, but the species measured by the mass spectrometer is the negatively charged molecule. And so the species they measure does not have a mass of 505.9879 but rather an m/z of 505.9885 because the additional electron has a mass of .00054858.
- iv) Figure 7. The m/z given for the glucose-6-phosphate [M-H]⁻ anion is not correct. The mass they have given is that of the [M-H] neutral molecule, but the species measured by the mass spectrometer is the negatively charged molecule. And so the species they measure does not have a mass of 259.0219 but rather an m/z of 259.0224 because the additional electron has a mass of .00054858.
- v) Figure 7G. It is not clear why the authors have assigned the data to glucose-6-phosphate when its isomer, fructose-6-phosphate, has identical mass.
- vi) Figure 7G. The images are interpolated and the color scales saturated. This is not accepted practice in the MSI field. The merged images (bottom row) do not add to the manuscript as the underlying histological images are barely visible. Please specify how the MSI and histological images co-registered, and provide mass accuracy for all assignments. Note: this reviewer is concerned how the authors can state in the methods section that “Metabolite identifications were made based on accurate mass, which typically has a discrepancy of less than 1 ppm” when the calculated masses referred to the figure captions are all incorrect (for instance the mass of an electron contributes approximately 2ppm mass error for glucose-6-phosphate).

vii) Figure 7G. The authors have based their analysis on a single circular ROI for the tumor and the adipocyte regions for each animal. However it is not clear how these ROI's were selected or if the results would change if different ROIs were selected. For instance the adipocyte ROIs indicated in Figure 7G for the 0h and 1h time points have clearly very different morphological characteristics). Please provide clear criteria used for ROI selection, and average multiple ROIs from each animal for both tumor and adipocyte regions.

viii) Please refer to the MetaSpace program for help with confident assignment of metabolites to MSI data.

Response to Referees # NCOMMS-19-01545B

Reviewer #2

Comment: The western blot results appear to be from single, non-replicated experiments (eg, Figs 3e, 5, SI5c) rather than representative of replicate independent experiments. If this is the case, the results are not sufficient for the mechanistic interpretations presented from these results. Please clarify / add experiments and statistical tests.

Response: The values presented in Figs 3e, 4f, 5, 6d; Supplementary Figs 4d, 5b, 5c, 7c, 8b are relative normalized protein levels with respect to the corresponding control for the Western blots shown. The experiments were repeated three times independently. The average values of the normalized protein levels and statistical analyses for the three experiments are presented in Supplementary Figs. 10 and 11 of the revised manuscript. The corresponding Figure Legends section were amended accordingly in the revised manuscript.

Comment: Many journals now also require the full blots in SI. Please check if this is required.

Response: Full blots of all western blots are supplied as Supplementary Figures 12 to 15.

Comment: LTF levels are graphed in SI Fig 2h and SI3b as U/mL. In line 196 it states that ovarian cancer cells with treated" ... with 100 ug/ml (a physiological level) of LTF in serum." I cannot find the conversion from Units to ug. Please include this in either the results or methods.

Response: Human LTF levels in sera shown in Supplementary Figs. 2h and 3b were measured using commercially available ELISA kit according to the manufacturer's protocol. From the results, the average LTF level in sera is around 250 U/mL while that in ascites is 450 U/mL. The values are presented as Unit per mL with unit as an arbitrary amount. There is not a conversion from unit to μg . On the other hand, for *in vitro* experiment, it was reported that 100 $\mu\text{g}/\text{mL}$ LTF promoted invasiveness in triple-negative breast cancer while LTF levels range from 10 $\mu\text{g}/\text{mL}$ in healthy individuals to 200 $\mu\text{g}/\text{mL}$ in individuals with inflammation.

Comment: Methods line 660-661 – “ ... IG10 cells were injected intraperitoneally into female C57BL/6 mice for 6 weeks ...” Please correct – cells should only be injected once.

Response: The Methods section was amended accordingly in the revised manuscript as follows, “To determine the effect of ovarian cancer cells on serum ITLN1 levels *in vivo*, mouse ovarian cancer IG10 cells (2×10^6) were injected intraperitoneally into female C57BL/6 mice at the age of 6 weeks old. Mice were euthanized using a carbon dioxide chamber followed by cervical dislocation after 6 weeks.”

Comment: Figure 7. Please include the timepoint shown for each experiment.

Response: Time points of each experiment for Fig. 7a to 7f were described in the Methods section under the “*In vivo* studies” subsection in the manuscript. They are also described in the corresponding Figure Legends section in the revised manuscript.

Comment: Was there an increase in survival for the mice treated with recombinant ITLN1?

Response: In this study, we focus on examining the effect of recombinant ITLN1 on ovarian cancer cell growth *in vivo*. We have only performed *in vivo* studies for a fixed period of time (six

weeks after recombinant ITLN1 treatment). We found that recombinant ITLN1 significantly suppressed ovarian cancer growth *in vivo* six weeks after treatment. However, we do not have the survival data for the mice treated with recombinant ITLN1.

Reviewer #4

Comment: Figure 7H and Supplementary Figures 8f and 8g. The figures claim to report the normalized abundances of glucose-6-phosphate, lactate and ATP. It is not explained how the data was normalized, as such it is not possible to understand what metric is being displayed in the MSI figures.

Response: Normalization of metabolite abundance was done in data analysis software Progenesis QI by default. It is required in differential experiments to calibrate data acquired across sample runs. This corrects for factors that result in experimental or technical variation introduced by the system when acquiring IMS data for over a long period of time. In brief, a ratiometric approach was used to select one run as the normalization reference to normalize the measured compound ion abundance to all compounds. Details are found in <http://www.nonlinear.com/progenesis/qi/>. The explanation of how the data was normalized is described in the Methods section under the “MALDI-IMS” subsection in the revised manuscript

Comment: Supplementary Figures 8f. The figure caption states “lactate abundance (ion adduct: [M-H]⁻; m/z: 88.01604).” However, the m/z is not correct. In atomic mass units the correct m/z of the [M-H]⁻ anion is 89.0244176 (neutral molecule is 90.03169406). It is not clear how the authors have assigned an incorrect mass to lactate.

Response: Thank you for the comment. The monoisotopic molecular weight of the compound of interest is obtained from the Human Metabolome Database (HMDB). The molecular weight of the ion adduct is calculated using online tool from University of California, Irvine (<https://www.lfd.uci.edu/~gohlke/molmass/>). The monoisotopic molecular weight of lactic acid (C₃H₆O₃) is 90.0317 and that of the [M-H]⁻ ion adduct (C₃H₅O₃) is 89.0239 (or 89.0244 with an extra electron). Due to limitation of the TOF mass spectrometer and even with mass calibrated to within 5ppm error prior to acquiring the IMS data, the difference in mass with or without an electron would not be distinguishable, which the measured mass error for these runs are typically around 10ppm. There was a mistake in the corresponding Methods section in the previous version of the manuscript. The discrepancy should be rated to less than 10 ppm instead of 1 ppm. The corresponding Methods and Figure Legends section is revised according to reviewer’s comment. The molecular weight 89.0239 was assigned to the ion adduct [M-H]⁻ for lactate when doing analysis. However, the mistake was made when writing the Figure Legends section, the molecular weight of an extra H was taken away and so “m/z: 88.01604” was resulted. The labels in Supplementary Fig. 8f are also corrected to “Normalized lactic acid abundance” in the revised manuscript for accuracy.

Comment: Supplementary Figures 8f. The m/z given for the ATP [M-H]⁻ anion is not correct. The mass they have given is that of the [M-H] neutral molecule, but the species measured by the mass spectrometer is the negatively charged molecule. And so the species they measure does not have a mass of 505.9879 but rather an m/z of 505.9885 because the additional electron has a mass of .00054858.

Response: The monoisotopic molecular weight of ATP (C₁₀H₁₆N₅O₁₃P₃) is 506.9957 and that of the [M-H] ion adduct (C₁₀H₁₅N₅O₁₃P₃) is 505.9879 (or 505.9885 with an extra electron). For the same reason mentioned in the previous response, due to limitation of the TOF mass spectrometer and even with mass calibrated to within 5ppm error prior to acquiring the IMS data, the difference in mass with or without an electron would not be distinguishable, which the measured mass error for these runs are typically around 10ppm. There was a mistake in the corresponding Methods section in the previous version of the manuscript. The discrepancy should be rated to less than 10 ppm instead of 1 ppm. The corresponding Methods and Figure Legends section is revised according to reviewer's comment.

Comment: Figure 7. The m/z given for the glucose-6-phosphate [M-H]⁻ anion is not correct. The mass they have given is that of the [M-H] neutral molecule, but the species measured by the mass spectrometer is the negatively charged molecule. And so the species they measure does not have a mass of 259.0219 but rather an m/z of 259.0224 because the additional electron has a mass of .00054858.

Response: The monoisotopic molecular weight of glucose-6-phosphate (C₆H₁₃O₉P) is 260.0297 and that of the [M-H] ion adduct (C₆H₁₂O₉P) is 259.0219 (or 259.0224 with an extra electron). For the same reason mentioned in the previous response, due to limitation of the TOF mass spectrometer and even with mass calibrated to within 5ppm error prior to acquiring the IMS data, the difference in mass with or without an electron would not be distinguishable, which the measured mass error for these runs are typically around 10ppm. There was a mistake in the corresponding Methods section in the previous version of the manuscript. The discrepancy should be rated to less than 10 ppm instead of 1 ppm. The corresponding Methods and Figure Legends section is revised according to reviewer's comment.

Comment: Figure 7G. It is not clear why the authors have assigned the data to glucose-6-phosphate when its isomer, fructose-6-phosphate, has identical mass.

Response: The labels in Figure 7g and h are corrected to "Normalized hexose-6-phosphate abundance" in the revised manuscript. The corresponding Results and Figure Legends sections are also revised. In this study, we aim at using the intermediate product of glycolysis as a measure of glucose consumption. Although we cannot distinguish glucose-6-phosphate from fructose-6-phosphate by their masses, the decreases in hexose-6-phosphate (glucose-6-phosphate and fructose-6-phosphate) in tumor cells and their increases in adipocytes adjacent to tumor cells after ITLN1 injection suggest that ITLN1 treatment decreases glucose uptake of metastatic OC cells but increases glucose consumption of neighboring adipocytes in the omental microenvironment.

Comment: Figure 7G. The images are interpolated and the color scales saturated. This is not accepted practice in the MSI field. The merged images (bottom row) do not add to the manuscript as the underlying histological images are barely visible. Please specify how the MSI and histological images co-registered, and provide mass accuracy for all assignments. Note: this reviewer is concerned how the authors can state in the methods section that "Metabolite identifications were made based on accurate mass, which typically has a discrepancy of less than 1 ppm" when the calculated masses referred to the figure captions are all incorrect (for instance the mass of an electron contributes approximately 2ppm mass error for glucose-6-phosphate).

Response: The heat map images were produced by High Definition Imaging software (HDI; Waters) when the tissue sections were scanned using the mass spectrometer. The interpolated images are for better visualization and the saturated color scale gives a better insight of the

distribution of compounds and assists in coregistering with the histological images. However, quantification of the abundance of the compound of interest is done using Progenesis QI by importing the ROI raw data and is independent of the visual of the images. The merged images are removed from Fig. 7g in the revised manuscript according to the reviewer's comment.

The same tissue section was washed and stained with hematoxylin and eosin (H&E) after data acquisition with the mass spectrometer. The image of H&E staining and heat map image were overlaid using HDI by the shape of the tissue section. This is described in the Method section under the "MALDI-IMS" subsection in the revised manuscript. All masses of compounds of interest are revised accordingly as well.

Comment: Figure 7G. The authors have based their analysis on a single circular ROI for the tumor and the adipocyte regions for each animal. However it is not clear how these ROI's were selected or if the results would change if different ROIs were selected. For instance the adipocyte ROIs indicated in Figure 7G for the 0h and 1h time points have clearly very different morphological characteristics). Please provide clear criteria used for ROI selection, and average multiple ROIs from each animal for both tumor and adipocyte regions.

Response: Fig. 7g shows representative images from MALDI-IMS of one ROI for tumor and adipocyte regions. Three ROIs for each region (tumor and adipocyte) were randomly selected based on histology of cell types for the analysis. Average of the three ROIs was taken before calculating for statistical significance. This is described in the Method section under the "MALDI-IMS" subsection in the revised manuscript. Adipocytes are identified as rounded cells with over 90% cell volume taken up by a single fat droplet. The ROIs were selected by pixel for analysis and the circles on the representative images in Fig. 7g are shown for better visualization purposes. They are revised in the current version of the manuscript for better representation of the ROIs.

Comment: Please refer to the MetaSpace program for help with confident assignment of metabolites to MSI data.

Response: We appreciate the reviewer's suggestion for using the MetaSpace program. We will consider using MetaSpace in a future opportunity as it does offer another method of validation but we believe the current method of using Progenesis QI for the engine is sufficient for the time being.

Reviewers' comments:

Reviewer #4 (Remarks to the Author):

The authors have attempted to address the comments this reviewer had concerning the MALDI MSI data, specifically with regard to assignments, normalisation, and selection of ROI's.

The normalisation of mass spectrometry data from different experiments is a critical factor when seeking to determine changes in metabolite levels. The answer to my question regarding how the MSI data were normalised, and the revised text, is essentially we used the Progenesis Q1 software in its default settings, please see the Progenesis software for an explanation. This is not sufficient. It is up to the authors to demonstrate their method is valid. Furthermore this reviewer has checked the Progenesis website (not exhaustively) and have found no explanation. As it is a TOF based system it is probably a TIC based normalisation, but the authors and readers need to understand better how their data is processed. Please provide details.

This reviewer is not particularly enamoured with manuscripts that change the criteria of their measurements substantially in a revision. From <1ppm for assignments in the original to <10ppm in the revision. Please specify how 10ppm mass accuracy was determined. It has been demonstrated many times that high mass accuracy is needed for metabolite MSI, and the risk of false positive ID's increases with decreasing mass accuracy. The methods section now simply states that "Masses were searched against the Human Metabolome Database for identifications" - please include all search criteria.

The Metaspacer program referred to previously by this reviewer, and then summarily (but politely) dismissed by the authors, was designed to improve confidence in metabolite assignments. Mass accuracy is one critical element, and also uses isotopic abundance etc... The low mass accuracy of the current data is probably insufficient for the Metaspacer program, which kind of highlights the potential uncertainty in the data. This reviewer would be much more confident with some independent validation of the observations (e.g. LCM followed by LC-MS/GC-MS).

Ultimately this reviewer still has several reservations about the MSI data. It may well be correct but it is vulnerable to false positive identifications which have then been interpreted in the best possible light. Some verification would sate these concerns.

Response to Referees # NCOMMS-19-01545C

Reviewer #4

Comment: The normalisation of mass spectrometry data from different experiments is a critical factor when seeking to determine changes in metabolite levels. The answer to my question regarding how the MSI data were normalised, and the revised text, is essentially we used the Progenesis QI software in its default settings, please see the Progenesis software for an explanation. This is not sufficient. It is up to the authors to demonstrate their method is valid. Furthermore this reviewer has checked the Progenesis website (not exhaustively) and have found no explanation. As it is a TOF based system it is probably a TIC based normalisation, but the authors and readers need to understand better how their data is processed. Please provide details.

Response: Normalization of metabolite abundance was done in data analysis software Progenesis QI using total ion count (TIC). In summary, Progenesis QI uses a median and mean absolute deviation approach based on all the detected abundance to calculate quantitative abundance ratios that models the differences in ion abundances between samples and apply an approximation of the mean and variance of the ratio distribution to finalize the calculation of the scaling factor. Further details are found in <http://www.nonlinear.com/progenesis/qi/v1.0/faq/how-normalisation-works.aspx>.

The explanation of how the data was normalized is described in the Methods section under the "MALDI-IMS" subsection in the revised manuscript.

Comment: This reviewer is not particularly enamoured with manuscripts that change the criteria of their measurements substantially in a revision. From <1ppm for assignments in the original to <10ppm in the revision. Please specify how 10ppm mass accuracy was determined. It has been demonstrated many times that high mass accuracy is needed for metabolite MSI, and the risk of false positive ID's increases with decreasing mass accuracy. The methods section now simply states that "Masses were searched against the Human Metabolome Database for identifications" - please include all search criteria.

Response: The mass accuracy specifications for the SYNAPT G2-Si high-resolution mass spectrometer is 3 ppm. However, during acquisition, the accuracy can be influenced by other factors, such as temperature changes. Although we calibrated the mass using red phosphorus prior to acquiring the data, the mass accuracy drifts during the acquisition. To determine the mass accuracy for this particular experiment, we calculated the mass error between the experimental and theoretical m/z for 9-aminoacridine (9-AA), the matrix we used for negative mode acquisition. The resulting calculation is < 10 ppm.

The search criteria are as follows, (i) mass resolution is 20000; (ii) pixel size is 60 x 60 μm ; (iii) ion adduct formed is [M-H]⁻ for negative mode and (iv) the database for metabolite annotation is Human Metabolome Database (HMDB). These are included in the Methods section under the "MALDI-IMS" subsection in the revised manuscript.

Comment: The Metaspacer program referred to previously by this reviewer, and then summarily (but politely) dismissed by the authors, was designed to improve confidence in metabolite assignments. Mass accuracy is one critical element, and also uses isotopic abundance etc... The low mass accuracy of the current data is probably insufficient for the Metaspacer program, which kind of highlights the potential uncertainty in the data. This reviewer would be much more confident with some independent validation of the observations (e.g. LCM followed by LC-MS/GC-MS).

Response: While it is not our intent, we apologize for appearing to regardlessly dismiss the suggestion. We considered that Metaspacer is optimized for data from high-resolving power mass spectrometry like FTICR and Orbitrap, which have mass accuracies in 1-5 ppm range (Spraggins et al., 2015 [<https://link.springer.com/article/10.1007/s13361-015-1147-5>]; Thermo Scientific MALDI LTQ Orbitrap [http://maldi-msi.org/wp/wp-content/uploads/2010/09/BR30159-MALDI-LTQ_Orbitrap.pdf]). Although the specifications for mass accuracy of our instrument is in similar range, the experimental mass accuracy is within 10 ppm under operational conditions as mentioned above. Moreover, the performance of Metaspacer was reported to change significantly at mass accuracy greater than 5 ppm (Palmer et al., 2017, Fig. 2i and Supp. Fig. 5 [<https://www.nature.com/articles/nmeth.4072>]). The performance was evaluated using dataset of ions with m/z in 100-1200 range, which is comparable to the range of metabolite ions that we have extracted.

REVIEWERS' COMMENTS:

Reviewer#4: (Remarks to the Author)

The authors have addressed this reviewer's concerns and this reviewer recommends publication. The interpretation of the MSI data is fully consistent with establish metabolite MSI observations. While one could claim (with some justification) that the mass accuracy is lower than that recommended, and that the study lacks independent verification, it is also true that MSI routinely detects the same set of metabolites, the identities of which are well established.